# Metatranscriptome Analysis of Sympatric Bee Species Identifies Bee Virus Variants and a New Virus, Andrena-Associated Bee Virus-1

**DOI:** 10.3390/v13020291

**Published:** 2021-02-12

**Authors:** Katie F. Daughenbaugh, Idan Kahnonitch, Charles C. Carey, Alexander J. McMenamin, Tanner Wiegand, Tal Erez, Naama Arkin, Brian Ross, Blake Wiedenheft, Asaf Sadeh, Nor Chejanovsky, Yael Mandelik, Michelle L. Flenniken

**Affiliations:** 1Department of Plant Sciences and Plant Pathology, Montana State University, Bozeman, MT 59717, USA; kdaughenbaugh@gmail.com (K.F.D.); brian.ross6@student.montana.edu (B.R.); 2Pollinator Health Center, Montana State University, Bozeman, MT 59717, USA; charlieccarey@gmail.com (C.C.C.); alexmcmenamin2@gmail.com (A.J.M.); tnrwgnd@gmail.com (T.W.); 3The Faculty of Agriculture, Food and Environment, The Hebrew University of Jerusalem, Rehovot 5290002, Israel; idan.kanonitch@mail.huji.ac.il (I.K.); yael.mandelik@mail.huji.ac.il (Y.M.); 4Agroecology Lab, Newe Ya’ar Research Center, ARO, Ramat Yishay 30095, Israel; naamaark@gmail.com (N.A.); asafsa@volcani.agri.gov.il (A.S.); 5Department of Microbiology and Immunology, Montana State University, Bozeman, MT 59717, USA; bwiedenheft@gmail.com; 6Entomology Department, ARO, The Volcani Center, Rishon Lezion 7528809, Israel; tal.erez2@mail.huji.ac.il (T.E.); ninar@volcani.agri.gov.il (N.C.); 7The Mina & Everard Goodman Faculty of Life Sciences, Bar Ilan University, Ramat Gan 5290002, Israel

**Keywords:** honey bee virus, *Apis mellifera*, *Andrena*, virus transmission, RNA virus, Andrena-associated bee virus-1 (AnBV-1), Lake Sinai virus, black queen cell virus, deformed wing virus, bees, pollinators, virus ecology

## Abstract

Bees are important plant pollinators in agricultural and natural ecosystems. High average annual losses of honey bee (*Apis mellifera*) colonies in some parts of the world, and regional population declines of some mining bee species (*Andrena spp*.), are attributed to multiple factors including habitat loss, lack of quality forage, insecticide exposure, and pathogens, including viruses. While research has primarily focused on viruses in honey bees, many of these viruses have a broad host range. It is therefore important to apply a community level approach in studying the epidemiology of bee viruses. We utilized high-throughput sequencing to evaluate viral diversity and viral sharing in sympatric, co-foraging bees in the context of habitat type. Variants of four common viruses (i.e., black queen cell virus, deformed wing virus, Lake Sinai virus 2, and Lake Sinai virus NE) were identified in honey bee and mining bee samples, and the high degree of nucleotide identity in the virus consensus sequences obtained from both taxa indicates virus sharing. We discovered a unique bipartite + ssRNA Tombo-like virus, Andrena-associated bee virus-1 (AnBV-1). AnBV-1 infects mining bees, honey bees, and primary honey bee pupal cells maintained in culture. AnBV-1 prevalence and abundance was greater in mining bees than in honey bees. Statistical modeling that examined the roles of ecological factors, including floral diversity and abundance, indicated that AnBV-1 infection prevalence in honey bees was greater in habitats with low floral diversity and abundance, and that interspecific virus transmission is strongly modulated by the floral community in the habitat. These results suggest that land management strategies that aim to enhance floral diversity and abundance may reduce AnBV-1 spread between co-foraging bees.

## 1. Introduction

Bees gather plant-produced pollen and nectar as their protein and carbohydrate sources, respectively, and in doing so provide a critical ecosystem service—the pollination of numerous plant species. Approximately 87.5% of flowering plants benefit from animal-mediated pollination, including bees [1]. Bee-pollinated plants include those that produce fruit, nut, and vegetable crops, with a global annual value of $175 billion dollars (USD) [2]. There are over 20,000 known bee species worldwide [3]. Bees differ in many life-history traits including diet breadth (i.e., specialist or generalist foragers based on their preference or lack thereof for particular plants); sociality, ranging from solitary to eusocial; nesting habits, including ground, cavity, and open nesting; activity season; body size and proboscis length [3,4,5], and related flight distance [6]. These traits influence their nutritional requirements, ability to adapt to land-use changes and climate change, and their susceptibility to pathogens.

In many parts of the world, insect populations have declined [7]. These losses are attributed to multiple factors including habitat destruction, insecticide exposure, the loss of quality nutritional resources, climate change, parasites, and pathogens [8,9,10,11]. In particular, many bee populations have suffered range reductions and local extinction, including the rusty patched bumble bee (*Bombus affinis*) and the western bumble bee (*Bombus occidentalis*) in the US [12,13,14]. The factors influencing bee population declines vary geographically. In some parts of the world, local bee populations have expanded in part due to management practices (e.g., an increased number of managed *Apis mellifera* colonies in Argentina and Spain), or have altered occupancy patterns due to climate change and/or urbanization [15,16,17]. Regardless of population trends, insects in general (>1 million species) and bees in particular remain greatly understudied [18].

Although the majority of bee taxa are under-researched, honey bees (*Apis mellifera*) are one of the most studied insects due to their wide global distribution and their importance in agriculture. Honey bees are eusocial insects that live in colonies of ~30,000 individuals. Colonies are composed of predominantly sterile female workers, one reproductive female (the queen), and hundreds of seasonal males (drones) [19]. Beekeepers manage honey bee colonies for honey production and pollination services. Honey bees forage across longer distances (i.e., typically within two kilometers, but up to 10 km from their hive) compared to most other bee species [20,21,22].

They are generalist foragers that readily visit and pollinate a wide variety of crops and wild flowers. These traits, coupled with the ability to transport managed hives to pollinate many crops, have increased agricultural reliance on migratory beekeeping operations. This is particularly true in the US, where large-scale monoculture is common practice, but also in other parts of the world, including Israel, which has one of the highest beekeeping densities in the world (i.e., 10,000 hives per 7,000 km^2^) [23]. High annual losses of managed honey bee colonies in many parts of the world (e.g., 37% average annual loss in the United States from 2010 to 2018 and up to 40% in Israel in 2008) [23,24,25,26,27,28,29,30,31,32,33] have heightened the scientific and public interest in the abiotic and biotic factors contributing to these losses and the ecological importance of all bee taxa, including native and/or wild bees such as mining bees.

Mining bees include over 1550 species in the genus *Andrena* (Andrenidae) [3,34]. These wild bees have a mostly Holarctic distribution [35] and are commonly found in both agricultural and wild landscapes. Bees in this genus are generally solitary and lack social structures, although some species are communal [3,36,37]. Unlike honey bees, all *Andrena* females can produce offspring, which they must provide nutritional resources, since there is no division of labor or overlap of generations [37]. *Andrena* species persist on a nectar and pollen diet and show a wide range of foraging behaviors from generalist (broadly polylectic, exploiting multiple plant families) to highly specialist (narrow oligolectic, foraging on a single plant genus) [38,39]. While adults of many species are usually only active for several weeks per year, oligolectic species time their emergence to coincide with the flowering period of preferred host plants [39]. In the Mediterranean and semi-arid eco-regions of Israel, *Andrena* species are prevalent [35].

High average annual losses of honey bee (*Apis mellifera*) colonies and regional population declines of some mining bee species (*Andrena spp*.) are attributed to multiple factors including habitat loss, lack of quality forage, insecticide exposure, and pathogens. Bee pathogens include bacteria, fungi, microsporidia, and numerous viruses [40,41,42,43]. The majority of research investigating the impact of viruses on bee losses has been carried out in honey bees, due to their importance as agricultural pollinators, their well-known biology and behavior, and the fact that they are managed and readily reared and manipulated at the individual and colony scale. Therefore, most bee-associated viruses are known as “honey bee viruses” in recognition of the host from which they were first discovered and described (reviewed in [42,44]). Most of the characterized bee-infecting viruses are positive-sense, single-stranded RNA viruses (+ssRNA) in the *Picornaviridae* family including Dicistroviruses (acute bee paralysis virus (ABPV), black queen cell virus (BQCV), Israeli acute paralysis virus (IAPV), and Kashmir bee virus (KBV)); the Iflaviruses (deformed wing virus (DWV), Kakugo virus, Varroa destructor virus-1/DWV-B, sacbrood virus (SBV), and slow bee paralysis virus (SBPV)); and taxonomically unclassified viruses (chronic bee paralysis virus (CBPV) and the Lake Sinai viruses (LSVs)) (reviewed in [42,45]). There are very few identified bee-infecting viruses with DNA genomes, which include Apis mellifera filamentous virus (AmFV) [46,47] and Osmia cornuta nudivirus [48,49]. Several honey bee-associated viruses have been detected in wild bee species including BQCV, LSV, SBV, and AmFV in *Andrena vaga*, and LSV and AmFV in *Andrena ventralis* in Belgium [50], and DWV, BQCV, and SBV in *Andrena spp*. sampled in the eastern United States [48,51,52,53,54,55,56]. It is unclear whether virus detections in most of these studies were indicative of active infections in mining bees since viral replication was not assessed by negative strand-specific RT-PCR or inferred by quantitative PCR, though DWV replication was detected in *Andrena haemorrhoa* in Germany [57].

Recent high-throughput sequencing studies identified additional honey bee-infecting +ssRNA viruses including Bee macula-like virus (BeeMLV) in the Tymoviridae family, Apis mellifera flavivirus, and Apis mellifera nora virus 1 [58,59], as well as the first negative-sense single-stranded RNA viruses (-ssRNA) including Apis mellifera rhabdovirus-1 (ARV-1) [59], also known as Bee rhabdovirus (BRV-1) [60] and Apis mellifera rhabdovirus-2 (ARV-2) (reviewed in [43]). To date, the majority of sequencing efforts aimed at identifying bee-associated viruses have utilized honey bee and the ectoparasitic mite (*Varroa destructor*) samples [48,59,60,61,62,63] (reviewed in [43]). Sequencing a greater variety of bee and other insect species indicates that many honey bee-infecting viruses have a broader host range that includes bumble bees (*Bombus spp*.), ants (*Camponotus spp.*) [64,65], and wasps (*Polistes fuscatus* and *Vespula vulgaris*) [56,64,66]. Schoonvaere et al. utilized high-throughput sequencing to detect previously characterized bee viruses and identify new virus and virus-like sequences associated with eight wild bee species, including three mining bee species—*Andrena cineraria*, *Andrena fulva*, *Andrena haemorrhoa*; three bumble bee species—*Bombus terrestris*, *Bombus cryptarum*, *Bombus pascuorum*; and two mason bee species—*Osmia bicornis*, and *Osmia cornuta*, collected from four locations in Belgium [49]. Overall, the normalized virus coverage data indicate that the majority of previously described bee viruses were not detected in this sample cohort. Likewise, incomplete genome coverage of detected viruses, including new putative virus-like sequences, suggests low virus abundance, though this result may be in part due to the use of polyA-based sequence libraries. For example, relatively few BQCV and DWV-B/VDV-1 reads were detected in *B. terrestris* and *B. cryptarum*, and *B. terrestris* and *O. cornuta*, respectively, suggesting either low/initial infection levels or detection of pollen-associated virus [49,51,56]. Furthermore, this study revealed that the viruses detected varied with sample site and bee taxa, indicating that viruses are not necessarily shared between co-foraging bee species. For example, Bee macula-like 2 virus was abundant in *A. haemorrhoa* samples, but absent from other bee samples, within a single site [49]. Identification of 11 new virus-like sequences in this study, including putative negeviruses, which were detected in numerous samples, suggests that additional sequencing studies are required to identify the complete suite of bee-infecting viruses. Moreover, studies that characterize bee-associated viruses are needed to confirm that virus-like sequences are bona fide viruses. Follow up studies that examine the prevalence and abundance of recently identified viruses in bees and other insects are also required to better understand the ecology of bee/insect-infecting viruses, including determining which viruses negatively impact bee communities.

Virus transmission within species occurs vertically from parents to offspring. Horizontal transmission within species occurs through several mechanisms including close contact, especially among eusocial colony-dwelling insect species and via behaviors such as trophallaxis (mouth-to-mouth food transfer) and grooming. Horizontal virus transmission, both within and between honey bee colonies, includes vector-mediated transmission (reviewed in [42,67]). For example, the *Varroa destructor* mites that parasitize honey bee colonies can be productively infected by several bee viruses (e.g., DWV [68,69,70,71], IAPV [72], KBV [73], and CBPV [74]). Viruses are transmitted to bees within and between honey bee colonies when infected mites feed on bees. Wax moths (*Galleria mellonella*) and small hive beetles (*Aethina tumida*) are other honey bee colony infesting insects that can host active IAPV and BQCV, and DWV infections, respectively [75,76,77,78,79]. It is likely that these insects host additional bee-infecting viruses. Carnivorous wasp and ant species can also be productively infected by DWV, IAPV, and BQCV [65,80]. Though commonly described as “bee virus” infections in other insects, virus transmission between parasitic insects and their hosts is bidirectional.

The most prevalent route of virus transmission between different bee taxa is likely via shared floral resources [53,56,81,82,83,84,85,86,87]. Specifically, infected bees can deposit viruses on flowers as they forage, and these viruses can then infect other bees that visit these flowers. Clear experimental based evidence of interspecies transmission was demonstrated in greenhouse studies of IAPV-transmission from honey bees to bumble bees and vice versa via shared food resource [56]. However, co-foraging species do not necessarily share viruses, as demonstrated by deposition of BQCV and DWV onto floral resources by honey bees, and lack of infection in bumble bees that foraged on the same flowers [53] and lack of virus infection via oral inoculation [88]. At the community level, interspecies virus transmission in sympatric honey bees and bumble bees was indicated by the extent of virus sequence identity in both taxa [87]. Directionality of infections is often inferred based on the prevalence and abundance of viruses in particular species within a sample cohort [52,87]. However, it is important to note that studies based on sample cohorts obtained at a single snapshot in time and/or from a single geographic location may not accurately represent virus ecology, as virus infections in honey bee colonies vary with sample date [89,90,91,92,93,94]. Thus, longitudinal studies with large samples sizes and/or the cumulative evidence from numerous smaller-scale studies are required to document biologically important trends. Examples of such studies include those that documented high DWV prevalence and abundance in the fall, coupled with high *Varroa* mite infestation correlate with over-winter honey bee colony death [69,89,91,95,96].

Evidence indicating that virologists, entomologists, and ecologists are in the initial phases of characterizing the bee-associated virome is provided in a recent study that sequenced virus-augmented samples from honey bees obtained as part of a large-scale sampling effort in Belgium, coupled with in-depth analyses of existing sequence data [97]. The study by Deboutte et al. not only resulted in the discovery of new virus-like sequences, virus strains, and potential recombinant viruses in honey bees, it also included bioinformatic analyses of over 5000 sequence datasets, which revealed that a high number of viral genomes were common to numerous Apidae family members, as well as widely distributed across different families within the order Hymenoptera [97]. These results, coupled with smaller-scale studies documenting virus replication in species beyond that of the host(s) from which particular viruses were discovered, suggest that many Hymenopteran viruses, and likely insect viruses in general, infect a wide range of species, and thus the concepts of species-specific viruses and paradigm of “spillover” from honey bees to other bee species should be reevaluated [97].

To evaluate virus diversity and prevalence in sympatric, co-foraging bee species, and to identify ecological factors that are associated with patterns of virus spread, we conducted a field survey across 14 sites in Israel within a single blooming season, sampling populations of honey bees and dominant mining bee species. From these samples, we identified bee-associated viruses using high-throughput sequencing, including viral genome variants of BQCV, DWV, LSV-2, and LSV-NE, which would likely escape detection with the use of widely adopted primer sets that were developed based on virus sequences obtained from North America and Europe. Sequencing efforts resulted in the discovery of a unique bipartite +ssRNA Tombo-like virus. This virus was very prevalent and abundant in *Andrena* samples and therefore named Andrena-associated bee virus-1 (AnBV-1). AnBV-1 was also detected in honey bee samples albeit at lower frequency and abundance. Statistical modeling indicated that the probability of AnBV-1 infection in honey bees was greater in habitats with low floral diversity and abundance, suggesting that interspecific virus transmission is strongly modulated by the floral community in the habitat. Laboratory based experiments were carried out to confirm that AnBV-1 is a bona fide virus that replicates in primary honey bee pupal cells maintained in culture. There is a dearth of bee viruses that can be studied in the lab, and therefore future development of AnBV-1 may result in a new experimentally tractable system for investigating bee host–virus interactions. These studies are essential to understanding the impact of viruses on bees.

## 2. Materials and Methods

### 2.1. Study Region and Survey Design

We conducted a field survey during March 2018 in the Judean Foothills, central Israel, a Mediterranean agro-ecosystem comprised of a mosaic of agricultural fields, planted forests, and shrublands. Honey bees are commonly managed in this region for crop pollination and honey production. Previous studies in this region revealed diverse and abundant wild bee communities in both agricultural fields and surrounding natural and semi-natural habitats [98,99]. The survey included a total of fourteen sites (Figure 1 and Appendix A); sites were patches of wild bloom dominated by yellow-flowering Brassicaceae species, mainly *Sinapis alba*, *Hirschfeldia incana*, and *Rapistrum rugosum*. The average distance between neighboring sites was 1914 m, ranging between 524 and 4960 m. Eight of the sites were characterized by up to six blooming floral species (that were seen visited by bees during the survey), and classified as low-floral-diversity habitats. The other six sites, classified as high-floral-diversity habitats, were characterized by having greater than nine blooming floral species (that were seen visited by bees during the survey), including families other than Brassicaceae. While this initial classification of the sites to high and low floral diversity was performed for the purpose of site selection, subsequent characterization of each site was based on pollinator resource use and floral abundance data recorded at the site (Figure 6).

Each site was sampled for one day, from 8:00 to 16:00. Sampling was conducted only during favorable weather conditions (temperature >17 °C, wind velocity <3 m/s, and clear or partially clear skies). In each site, we marked a 25 by 25 m plot where sampling of bee foraging activity and of the floral community was carried out. The bee foraging activity sampling was conducted by slowly walking throughout the plot and netting bees from flowers for 40 min (excluding bee handling time). Each captured bee was identified in situ to the lowest taxonomic level possible, immediately placed in a separate vial, and kept on dry ice. The visited flower species of each captured bee was recorded. At the end of each sampling day, all the collected specimens were transported to the lab and stored at −80 °C.

To sample the floral community in each site, we sampled each plot by randomly placing ten 1 m radius hoops. In each hoop, we identified all the flowering species and counted the abundance of each species’ distinct floral units. We defined distinct floral units for most taxa as individual flowers or distinct florets within sparse inflorescences. In Compositae, Umbelliferae and *Trifolium*, floral units were defined as individual inflorescences. To account for the large surfaces of inflorescences in these families, we multiplied their counts by their surface size relative to that of dominant single flowers in the field.

The studied region is generally characterized by a patchy, fine-grained land-use mosaic, in which wild flower patches are relatively similar to each other in terms of the floral community (verified visually during site selection). Therefore, each of our sampled plots well represented the flower patches in the broader area, in a radius of at least several hundreds of meters.

#### Virus Analysis

The two most abundant bee taxa in this sample cohort were mining bees (i.e., genus *Andrena*; *A. combusta ochraceohirta*, *A. aerinifrons levantina, A. urfanella* and an unidentified morphospecies of the subgenus *Truncandrena*) and honey bees (*Apis mellifera*). Therefore, virus identification efforts were focused on these two bee taxa. A subset of the samples (i.e., between 7 and 11 individuals, with a median number of 11 individuals) per taxa per site were used for virus identification via high-throughput sequencing, and subsequent determination of the prevalence and abundance of AnBV-1. The first round of sequencing was carried out on sequencing libraries representing each bee species (i.e., pooled *Apis mellifera* or *Andrena spp.* samples) obtained from the following sites: Galon3—low floral diversity, low relative *A. mellifera* activity; N. Zanoah—high floral diversity, high relative *A. mellifera* activity; R.B.Sh—low floral diversity, high relative *A. mellifera* activity; and Berko2—high floral diversity, low relative *A. mellifera* activity (relative honey bee activity per site was categorized as high/low for sites with more/less than 20% honeybee foragers, out of the total number of sampled bees, respectively). In addition, sequencing libraries were prepared from two virus-augmented samples, as described below. To identify the most prevalent and abundant viruses in additional sites from this study, a second round of sequencing was carried out on pooled virus-augmented samples from each species (i.e., *Apis mellifera* or *Andrena spp*.) obtained from the following sites: Nahala—low floral diversity, low relative *A. mellifera* activity; Alon—high floral diversity, high relative *A. mellifera* activity; R.B.Sh2—low floral diversity, high relative *A. mellifera* activity; and Alon2—high floral diversity, low relative *A. mellifera* activity.

### 2.2. RNA Extraction

Individual bees in 2 mL safe-lock Eppendorf tubes were homogenized in 500 µL sterile buffer (100 mM Tris, 100 mM NaCl Tris pH 7.6) using 2 tungsten balls (2.85 mm, Ultraparts) in a Geno/Grinder (Spex SamplePrep) at 1550 rpm for 1.5 min. Samples were centrifuged for 2 min at 13,500 rpm at 4 °C to pellet debris. Supernatants (i.e., 150–200 µL for *Andrena spp.* and 400 µL for *Apis mellifera*) were transferred to a new tube and combined with 1–2 volumes of Trizol reagent (Life Technologies), vortexed, and incubated at room temperature for 5 min. Next, 150 µL of chloroform was added and each sample was mixed by hand for 15 s, followed by incubation at room temperature for 3 min. Samples were centrifuged for 15 min at 12,000× *g* at 4 °C, and the aqueous phase was removed from each sample and transferred to a fresh tube. An equal volume of isopropanol was added to each sample, along with 20 µg glycogen to aid in precipitation. Samples were precipitated at −20 °C for 24–72 h, then centrifuged for 10 min at 12,000× *g* at 4 °C. Supernatants were carefully removed by pipetting, and each pellet was washed twice with 75% ethanol. Final pellets were air dried at room temperature for 15 min, then dissolved in 20–30 µL sterile distilled water. RNA was quantified by spectrometry using a NanoDrop instrument (ThermoFisher). RNA samples of poor quality, where A260/A280 <1.7, were re-precipitated with lithium chloride or sodium acetate by combining the sample with an equal volume of isopropanol containing 700 mM LiCl or 1.5 M NaOAc, and precipitated at −20 °C for 1.5 h. RNA was washed with 75% ethanol as before and suspended in sterile distilled water. All RNA samples were stored at −80 °C until analysis.

### 2.3. RNAseq Library Preparation and Sequencing

RNA was sent to the Roy J. Carver Biotechnology Center at the University of Illinois for library preparation (Illumina TruSeq Stranded RNA Sample Prep kit) and paired-end sequencing on Illumina HiSeq 4000 or MiSeq sequencing systems. Libraries were prepared and quantitated using an Illumina Library quantification kit (Kapa). Paired end sequencing libraries were generated according to manufacturers’ instructions (lllumina). In brief, RNA from individual bees (*n* = 11 per library) was pooled to generate sequencing libraries representing each bee species (i.e., *Apis mellifera* or *Andrena spp.*) obtained from four sites (i.e., Galon3, N. Zanoah, RBSh, and Berko2), resulting in a total of eight sequencing libraries. The goal of sequencing these eight samples was to sequence bee-associated viruses (genomes/transcripts). Therefore, ribosomal RNA (rRNA) was depleted using the Ribo-Zero-Gold (Human/Mouse/Rat), which is compatible with *Apis mellifera* rRNA and *Andrena spp*. rRNA, which is very similar to honey bee rRNA (92% identical, e = 0), prior to the preparation of the libraries. Each of the libraries were paired-end sequenced on a HiSeq 4000 using a HiSeq 4000 SBS sequencing kit version 1, yielding ~44 million reads (2 × 100 nt) per sample.

In addition, the first round of sequencing included two virus-augmented libraries (i.e., one library representing *Apis mellifera* (*n* = 44) and one library representing *Andrena spp.* (*n* = 44), pooled from the four sites mentioned above) were prepared as follows. To enhance the amount of viral RNA sequenced, bee lysates were augmented for virion-protected/encapsidated nuclease treated by mixing 150 µL bee lysate with 50 U Benzoase (Sigma) and 40 U RNase I (Thermo) in total reaction volume of 250 µL, incubated at 37 °C for 1.5 h, then extracted with Trizol reagent (Thermo) according to the manufacturer’s instructions. Glycogen (Thermo) was added at a concentration of 20 µg per sample to aid in nucleic acid precipitation. Illumina paired-end 100 bp libraries were prepared without polyA selection (since not all virus genome segments are poly-adenylated) or rRNA removal steps since presumably the majority of rRNAs and the cytoplasmic mRNAs would have been degraded by RNAse treatments. RNA quality and abundance were estimated by NanoDrop and Qubit, which indicated very low abundance of RNA in virus-augmented samples. Reverse transcription was primed using random hexamers, followed by second strand synthesis, and amplification. Sequencing was as described above (HiSeq 4000) except that these virus-augmented libraries were diluted such that they would account for only 5% of the total sequencing lane capacity. Sequencing was on a single HiSeq 4000 lane, in combination with the above mentioned eight samples and yielded ~12 million reads (2 × 100 nt) for each of these virus-augmented samples.

A second round of virus-augmented RNAseq was carried out similarly to the first round of virus-augmented samples, but with new pools for *Apis mellifera* (*n* = 44) and *Andrena spp*. (*n* = 44) from samples obtained from four additional sites (i.e., Alon, Alon2, RBSh, and Nahala). This second round was sequenced on an Illumina MiSeq and yielded ~3.5 million reads (2 × 250 nt) each.

To obtain additional sequence data for AnBV-1, lysate from a single mining bee with a high abundance of AnBV-1 (i.e., sample 13, Appendix A), was nuclease treated by mixing 150 µL bee lysate with 50 U Benzoase (Sigma) and 40 U RNase I (Thermo) in total reaction volume of 250 µL, incubated at 37 °C for 1.5 h, then extracted with Trizol reagent (Life Technologies) according to the manufacturer’s instructions. Glycogen (Thermo) was added at a concentration of 20 µg to aid in precipitation. The RNA from this sample was prepared and sequenced identically to the round two virus-augmented samples. This individual sample was sequenced on an Illumina MiSeq and yielded ~3 million reads (2 × 250 nt).

### 2.4. Apis mellifera (Honey Bee) Genome Alignment

To remove as much bee genome sequence from the RNAseq libraries prior to analyses, sequences were aligned to the *Apis mellifera* genome (i.e, GCA_003254395.2_Amel_HAv3.1_genomic.fna, downloaded from NCBI 2019–02-07).

### 2.5. HoloBee Database Expansion (March 2019)

To identify and group sequences in the RNAseq libraries that corresponded to previously sequenced honey bee-associated microbes and viruses, the HoloBee-barcode and HoloBee-mop databases were downloaded from (https://data.nal.usda.gov/dataset/holobee-database-v20161) and split into virus and non-virus portions [100]. The virus database was supplemented to reflect numerous discoveries of bee-sample associated putative viral sequences (Appendix A) [48,49,59,60,66]. Reads and contigs were also aligned (HISAT2) and queried (BLAST, blastn, and dc-megablast) against the ref_viruses_rep_genomes database (11,015 sequences, downloaded from NCBI as a BLAST database 2019–03–24) and CLC de novo viral binned contigs from eight wild bee species [49] (567 contigs, fig share, https://figshare.com/authors/Karel_Schoonvaere/4182034, downloaded 2019–02–01); Appendix A).

### 2.6. RNA Sequencing and Analyses

RNAseq and viral sequencing resulted in a total of 379,840,320, 2 × 100 nt read pairs, and 10,222,370, 2 × 250 nt read pairs (see Appendix A). Reads were downloaded from DNA Services at University of Illinois at Urbana-Champaign. The reads were already trimmed of their 3′ adaptor sequences and barcode sequences that distinguished samples within the same lane. FastQC was used to view qualities both before and after further quality trimming with Trimmomatic-0.36 [101]. An example trimming reads on a macbook: *java -jar /Applications/Trimmomatic-0.36/trimmomatic-0.36.jar PE -threads 8 read1.fastq.gz read2.fastq.gz read1.paired.fastq.gz read1.unpaired.fastq.gz read2.paired.fastq.gz read2.unpaired.fastq.gz ILLUMINACLIP:TruSeq3-PE-2.fa:2:10:10:6 LEADING:3 TRAILING:3 SLIDINGWINDOW:4:15 MINLEN:36*.

Reads surviving trimming as pairs were aligned to the honey bee genome using HISAT2 [102]. The HISAT2 indexes were built: *hisat2–2.1.0/hisat2-build “$target.fasta”*, where *“$target.fasta”* referred to the honey bee genome. To remove reads aligning to, for example, honey bee, HISAT2 was combined with gsed and samtools (1.9) [102,103] to recover only the unaligned reads: *hisat2-2.1.0/hisat2 –threads 6 –summary-file metrics.metrics -x “$target” -1 read1.fastq.gz -2 read2.fastq.gz | gsed ‘/\t\(77\|141\)\t[*]/!d′ | samtools sort -n –threads 4 | samtools fastq –threads 4 -1 new.R1.fastq -2 new.R2.fastq* -, where *“$target”* was the index built above (e.g., the HISAT2 index for honey bee). Reads that aligned to honey bee were thereby omitted from further analyses. If only one of the read mats aligned to honey bee, both mates were omitted.

A complete *Andrena* genome was not available and thus sequences obtained from *Andrena* samples were also aligned to the honey bee genome (to remove at least some *Andrena* reads). The number of reads for each sample that aligned to the honey bee genome varied for each sample, but averaged ~54% for honey bee samples and ~14% for *Andrena* samples (Appendix A).

The RNAseq library reads (sequences) that did not align to honey bee (i.e., neither mate in the pair aligned) were then aligned to non-viral sequences from the Holobee database (described above) and the *Lotmaria passim* (formerly known as *Crithidia mellificae*, sf) to identify and remove non-viral reads that aligned to sequences within the augmented honey bee holobiome [104]. The procedure was identical to removal of the honey bee aligned reads.

The remaining RNAseq paired-end reads (i.e., non-*Apis mellifera*, and with the holobee non-virus reads likewise removed) from each library were assembled into contigs using Trinityrnaseq v2.10.0 via a docker image (docker pull trinityrnaseq/trinityrnaseq, 2019 and 2020) on either a macbook pro or AWS EC2 instances (e.g., m5 d.12× large with 48 GiB Memory, 192 vCPU). Libraries from each sample were separately assembled. Contig assessment and quantification were performed using scripts included with trinityrnaseq. An example running trinityrnaseq via docker on EC2: *sudo docker run --rm -v‘pwd‘:‘pwd‘ trinityrnaseq/trinityrnaseq Trinity --seqType fq --CPU 46 --max_memory 182 G --left ‘pwd‘/read1.fastq --right ‘pwd‘/read2.fastq --output ‘pwd‘/out_trinity*.

For contig quantitation, we used the trinityrnaseq utility script, align_and_estimate_abundance.pl, specifying kallisto as the aligner [105]. An example on a macbook: *trinityrnaseq/util/align_and_estimate_abundance.pl --transcripts contigs.fasta --seqType fq --left read1.fastq.gz --right read2.fastq.gz --est_method kallisto --trinity_mode --prep_reference --output_dir out_quant*. At times, we added the parameter *--kmer-size = 21* to better align more variant reads. Kmer size of 21 was experimentally determined to balance the alignment of the most reads to viral transcripts, while minimizing alignment of possibly non-target reads. To create virus-level abundance (e.g., all BQCV), we added a mapping from all contigs to all virus groupings (as labeled by their BLAST results) and supplied that as a custom trans_map as an argument to *align_and_estimate_abundance.pl* using the *--gene_trans_map* option. In other words, where this script usually quantitates contigs on a trinity concept of gene, consolidating all the isoforms, we applied an organism name instead of gene, so that we could review summaries at the organism level.

To identify candidate viral contigs, the assembled contigs were aligned to NCBI’s BLAST nt database (downloaded 2020–05–26) and BLAST databases constructed from the candidate virus contigs of eight wild bee species using blastn and dc-megablast with NCBI’s BLAST+ suite (versions 2.10.1) [49,106]. Likewise, DIAMOND (version 0.9.35 downloaded 2020–06–21) was used to align contigs to nr with taxonomy enabled (nr, prot.accession2taxid.gz and taxdmp.zip downloaded 2020–05–26) [107]. To build the DIAMOND database (database version 3): *diamond makedb --in nr.gz --taxonmap prot.accession2taxid.gz --taxonnodes taxdmp/nodes.dmp --taxonnames taxdmp/names.dmp --db n*. The DIAMOND preconfigured reporting fields were extended to include stitle, sscinames, staxids, sphylums, skingdoms. An example aligning contigs to nr with DIAMOND on an EC2 instance (48 CPUs, 192 GiB memory, and 900 GiB NVMe SSD Drives i.e., EC2 m5 d.12 xlarge instance): *sudo diamond blastx --query contigs.fasta --db nr --sensitive --evalue 0.001 --max-target-seqs 1 --max-hsps 1 --index-chunks 1 --block-size 7 --log --outfmt 6 qseqid sseqid pident length mismatch gapopen qstart qend sstart send evalue bitscore stitle sscinames staxids sphylums skingdoms -o diamond_out.tsv*. BLAST results were post-processed, to retain the top hit of any putative viral contig, provided that the viral contig was greater than or equal to 400 nt.

To quantify reads and create coverage maps of the viral genome sequence variants described herein, reads were aligned and counted using kallisto and resulting bam files manipulated with using Samtools [103,105]. More specifically, reads were aligned (using a kallisto index built with --kmer-size = 21, appropriate for more sensitive alignments in this dataset) to all the candidate viral contigs to estimate abundances. For example, to align as many reads as possible to individual contigs, such as AnBV-1 RNA1 or other selected contigs, we created kallisto indexes from fasta files containing single contigs in parallel *parallel kallisto index --index = {/.}21kmer.kallisto_index --kmer-size = 21 {} 2> kallisto_index.log ::: *.fasta* (“{}” are placeholders for substitutions in parallel based on the glob “*” in *.fasta). Using those indexes, we aligned each sample to each individual contig to quantitate and to generate pseudobams for coverage maps, for example: *kallisto quant --threads = 6 --pseudobam --output-dir index_name.sample_name --index kallisto_index_of_contig read1.fastq.gz read2.fastq.gz*. Pseudobams were sorted and indexed using samtools. Read coverage was visualized by loading the indexed bam files into Integrative Genomics Viewer (IGV_MacApp_2.8.12_WithJava). RNAseq, Trinity assembly and BLAST analyses were performed on a Mid 2015 MacBook Pro (Quad-Core Intel Core I7, macOS 10.14–10.15) or high CPU and/or high Memory AWS EC2 instances (detailed above) with Amazon Linux AMIs. Job control, read counts, extracting, and relabeling candidate viral trinity contigs, etc., were performed using bash and R. Some examples within this section are simplifications. Detailed code is available at: https://github.com/charlieccarey/viral_transcriptome_of_bees.

### 2.7. Reverse Transcription/cDNA Synthesis

Reverse transcription (RT) reactions to produce complementary DNA (cDNA) were performed by incubating 300 ng of RNA from each RNA pool or individual RNA sample with M-MLV reverse transcriptase (Promega) and 500 ng random hexamer primers (IDT) for 1 hr at 37 °C according to the manufacturer’s instructions. Samples were diluted 1:1 with sterile distilled water prior to analysis.

### 2.8. Polymerase Chain Reaction (PCR)

PCR was performed according to standard methods [91,108]. In brief, 2 μL cDNA template was combined with 10 pmol of each forward and reverse primer, and amplified with 0.5 μL (5 U/μL) ChoiceTaq polymerase (Thomas Scientific) according to the manufacturer’s instructions using the following cycling conditions: 95 °C for 5 min; 95 °C for 30 s, 55 °C for 30 s, 72 °C for 30 s–1 min, 35 cycles; followed by final elongation at 72 °C for 4 min. PCR products were visualized by 2% agarose gel electrophoresis and staining with SYBRsafe (Invitrogen). Positive and negative control reactions were included for all analyses and exhibited the expected results. Select products were purified with the Qiaquick PCR Purification Kit (Qiagen), quantified by NanoDrop spectrometry, and Sanger sequenced.

### 2.9. Quantitative Polymerase Chain Reaction (qPCR)

Quantitative PCR (qPCR) using primer sets listed in Appendix A was utilized to quantify the viral RNA (i.e., genome and transcript) abundance in the samples. The specificity of virus-specific primer sets was confirmed through melt-point analysis, gel electrophoresis, and Sanger sequencing of select qPCR products. Primers for LSV (reverse primer only), BQCV, and AnBV-1 were designed based on the sequencing information obtained as part of this study. Importantly, qPCR analyses using previously described primers would have resulted in false-negatives since the viral consensus sequences described herein differed from previously reported virus strains.

All qPCR reactions were performed in triplicate using 2 μL cDNA template (diluted 1:1 with sterile distilled water) per reaction. In addition, each 20 μL qPCR reaction contained 1× ChoiceTaq Mastermix (Thomas Scientific), 0.4 μM each forward and reverse primer, 1× SYBR green (Life Technologies), and 3 mM MgCl_2_. Reactions were carried out in 96-well plates using a CFX Connect Real-Time instrument using Maestro software (Bio-Rad) with the following thermo-profile: pre-incubation at 95 °C for one minute; 40 cycles of 95 °C for 10 s, 60 °C for 20 s, and 72 °C for 15 s; final extension at 72 °C for 1 min, followed by melt curve analysis. Positive and negative control reactions were included for all qPCR analyses and exhibited the expected results.

Virus-specific qPCR-target amplicons were cloned into the pGEM-T (Promega) or pCR2.1TOPO (Life Technologies) vectors, as described previously [89]. Plasmid standards, containing 10^3^ to 10^9^ copies per reaction, were used as qPCR templates to assess primer efficiency and generate standard curves used for quantifying viral RNA copy numbers for each sample. Primer efficiencies were evaluated using qPCR assays of cDNA and plasmid dilution series, and calculated by plotting log_10_ of the concentration versus the crossing point threshold (C(t)) values and using the primer efficiency equation, (10^(1/Slope)-1)^ × 100) [109]. The virus-specific qPCR primer sets utilized in this study had efficiencies ranging from 86 to 103% and provided accurate quantitative assessment of >10^3^ RNA copies per reaction. The linear standard equations for virus-specific plasmid standards and qPCR primers listed in Appendix A were as follows: Rpl8 Ct = -3.937× + 40.506, R^2^ = 0.9982; LSV Ct = -3.252× + 39.697, R^2^ = 0.9979; DWV Ct = -3.4093× + 41.641, R^2^ = 0.99756; AmFv Ct = -3.3185× + 39.489, R^2^ = 0.99151; BQCV Ct = -3.2918× + 40.332, R^2^ = 0.99378; SBV Ct = -3.725× + 44.017, R^2^ = 0.99018; AnBV-1 RNA 1 Ct = -3.5923× + 42.246, R^2^ = 0.99893; AnBV-1 RNA2 Ct = -3.3896× + 44.259, R^2^ = 0.94813; negative strand AnBV-1 RNA1 Ct = -3.417× + 40.427, R^2^ = 0.9995; negative strand AnBV-1 RNA2 Ct = -3.5973× + 44.846, R^2^ = 0.9972. All reported qPCR data were normalized to genome copies per 100 ng RNA.

### 2.10. Statistical Analysis of qPCR

To test for differences in AnBV-1 RNA copies relative to the zero hour time point in honey bee pupal cells, base R version 3.3.3 was used to perform a pairwise Wilcoxon Rank Sums test with no multiple comparisons correction [110]. A Benjamini-Hochberg correction was performed to correct for increased rate of false positives for additional pairwise comparisons to the zero hour time point. Briefly, the *p*-values were ranked from lowest to highest, then each individual *p*-value’s Benjamini–Hochberg critical value was calculated with the formula (i/m) 0.25, where “I” is the *p*-value’s rank, m is the number of comparisons, and 0.25 is the false discovery rate. The largest *p*-value that was smaller than the critical value was considered as the cutoff for significance. Therefore, any *p*-value equal to or smaller than the cutoff was considered statistically significant and is indicated in each figure legend.

### 2.11. Negative Strand-Specific PCR

AnBV-1-positive samples were analyzed for the presence of negative-strand RNA1 and RNA 2 using strand-specific RT-PCR according to published methods [111]. Briefly, RNA from select samples was extracted with Trizol as above, and cDNA synthesis reactions were performed with MMLV-RT (Promega) according to the manufacturer’s instructions using negative strand-specific RNA 1 and RNA 2 primers (TAGS-AnBV-1-RNA1 and TAGS-AnBV-1-RNA2, respectively) tagged with an additional 21 nt of sequence at their 5′ end. The TAGS sequence (5′GGCCGTCATGGTGGCGAATAA3′) shares no homology with AnBV-1 nor to the honey bee genome. In brief, 50 ng RNA, relevant primer, and 0.5 mM each dNTP were combined with RT Buffer containing 200 U MMLV-RT (Promega), and 40 U RNaseOUT (Life Technologies). Reactions were incubated for 120 min at 37 °C, then unincorporated primers present in the RT reactions were digested with 2 U exonuclease I (Life Technologies) per reaction at 37 °C for 30 min, followed by heat inactivation at 85 °C for 5 min. PCR was performed using 2 μL of this cDNA template in 25 μL reactions containing 10 pmol each of a tag-specific forward primer (TAGS) and an RNA 1 or RNA 2 reverse primer (AnBV-1-RNA1-R545 and AnBV-1-RNA2-R659, respectively) using the following cycling conditions: 95 °C for 5 min; 95 °C for 30 s, 55 °C for 30 s, 72 °C for 30 s, 35 cycles; final elongation 72 °C for 4 min, hold at 12 °C. For negative controls, PCR was performed using template incubated in the absence of RT enzyme during the reverse transcription reaction, and with the reverse primer only. Positive controls included PCR with PCR primers (AnBV-1-RNA1-F366 and AnBV-1-RNA1-R545, and AnBV-1-RNA2-F461 and AnBV-1-RNA2-R659) for detection of RNA 1 and RNA 2, respectively. Self-priming was tested by performing reverse transcription reactions in the absence of exogenous primers, followed by PCR with qPCR primers as described above. PCR products were analyzed by 2% agarose gel electrophoresis and were stained with SYBR safe (Life Technologies).

### 2.12. Rapid Amplification of cDNA Ends

Rapid Amplification of cDNA Ends (RACE) was performed to extend the sequence of AnBV-1 RNA 1 and RNA 2 using the SMARTer RACE 5′/3′ kit (Takara) according to the manufacturer’s instructions. Briefly, RNA containing high amounts of AnBV-1 as determined by qPCR extracted from an individual *Andrena spp.* sample was used as template to make 5′ and 3′ RACE-ready cDNA products. Prior to making 3′ RACE ready cDNA approximately 500 ng of RNA was polyadenylated using *E. Coli* Poly(A) Polymerase (New England Biolabs) according to the manufacturer’s instructions, since RNAseq data indicated that neither RNA 1 nor RNA 2 were polyadenylated. PCR for the 5′ end of RNA 1 was performed using 5′RACE-ready cDNA as template and primer AnBV-1-RNA1-R73. This PCR product was used as template for nested PCR using primer AnBV-1-RNA1-R43. The resulting PCR product was cloned using the In-Fusion system supplied with the RACE kit. Approximately 10–12 colonies were selected and grown in Leuria-Bertani broth supplemented with 100 μg/mL ampicillin and cultured overnight at 37 °C. Plasmid DNA was extracted using the Promega PureYield Plasmid Miniprep System (Promega). Plasmid DNA was sequenced using the M13R sequencing primer, and sequences were aligned to the corresponding consensus sequence obtained by RNAseq. PCR for the 3′ end of RNA 1 was performed using 3′RACE-ready cDNA as template and primer AnBV-1-RNA1-F1862. This PCR product was used as template for nested PCR using primer AnBV-1-RNA1-F1972. PCR for the 5′ end of RNA 2 was performed using 5′RACE-ready cDNA as template and primer AnBV-1-RNA2-R122. This PCR product was used as template for nested PCR using primer AnBV-1-RNA2-R69. PCR for the 3′ end of RNA 2 was performed using 3′RACE-ready cDNA as template and primer AnBV-1-RNA2-F2572. This PCR product was used as template for nested PCR using primer AnBV-1-RNA2-F2612. All RACE products were cloned, sequenced, and aligned as described above.

### 2.13. Honey Bee Pupal Cell Cultures and AnBV-1 Infection

Purple-eyed honey bee pupae were removed from the wax comb cells, in which they develop, using featherweight forceps (Bioquip). The pupae were incubated at 28 °C overnight in 12-well plates. Damaged (i.e., melanized) pupae were discarded. Primary cells were harvested from surface sterilized honey bee pupae in a biosafety cabinet. Surface sterilization was carried out in a 50 mL conical tube in which pupae were swirled in 0.6% hypochlorite solution (diluted bleach) for 3 min, 70% ethanol for 3 min, and in sterile water for injection (Gibco). In groups of two, pupae were dissected into head, thorax, and abdomen segments in 2 mL WH2 medium [112] in a 47 mm dish using sterile 18-gauge needles to vigorously disturb tissues and release cells into the medium. The cells were transferred to a 15 mL conical tube, while the carcasses were washed again with 2 mL of fresh medium, which was added to the same conical tube. Carcasses were then discarded. The total volume was brought to 14.5 mL with fresh WH2 medium, and 300 μL of this cell suspension were plated into each well of a 48 well plate (approximately 10^6^ cells per well) and incubated at room temperature overnight before infection.

To infect cells with AnBV-1, 3 μL *Andrea spp*. lysate containing 1.3–2.6 × 10^8^ copies AnBV-1 per µL, or lysate without virus (mock), were added to each well of pupal cells and incubated at room temperature. To quantify virus abundance over the course of infection, RNA was isolated from cell culture wells on multiple days post-infection (dpi) (i.e., 0, 1, 2, 3, 4, 5, or 7 dpi) using two volumes of Trizol reagent (Life Technologies) (i.e., 600 μL per well) and pipetting to dislodge adherent cells and facilitate cell lysis. The entire contents of each well were transferred into 1.5 mL centrifuge tubes and RNA was extracted according to manufacturers’ instructions as described above.

### 2.14. Andrena-associated bee virus-1 (AnBV-1) Genome and Phylogenetic Analyses

AnBV-1 open reading frames were predicted in SnapGene Viewer (www.snapgene.com) using the standard start codon option and using the Glimmer ORF finder tool in Geneious Prime (version 2020.2.3, www.geneious.com). Predicted ORF sequences and their translations were queried with BLAST and PHMMER online utilities [113,114]. AnBV-1 ORF1 is 1740 nt and encodes a putative 540 aa protein. Initial BLAST analyses (i.e., protein-protein BLAST (BLASTp) and Position-Specific Iterated BLAST (PSI BLAST) did not identify orthologous proteins and/or motifs in ORF1, ORF3, or ORF4. However, subsequent PHMMER analyses indicated that ORF2 of RNA1 encodes a 161 amino acid (aa) virion protein based on its similarity to a putative virion protein from spider-associated Loderio virus (NC_031748) [115]. AnBV-1 RNA 1 ORF2 shares 25.7% aa identity with Loderio virus ORF3, which is a 193 aa protein in the European Molecular Biology Laboratory database (EMBL-EBI), and both of these proteins fall into the SP24 (PF16504) family (Uniprot A0A1D9CFP5).

To construct a maximum likelihood phylogeny of viral RNA-dependent RNA polymerases (RdRp), 57 protein or polyprotein sequences were selected from a published list of insect and plant infecting viruses and were downloaded from NCBI (access data October 2020) [49]. These sequences were aligned using the LINSI option in MAFFT [116]. Columns composed of ≥80% gaps were removed from the alignment with Trimal before the alignment was manually inspected to remove sequences composed primarily of gapped positions [117]. ProtTest 3 was used to determine the best model for phylogenetic tree building and the alignment was used to construct a tree in PhyML with the BLOSUM62 + I + G model [118,119]. The resulting tree was rendered in FigTree [120]. GenBank accession numbers for either the RdRp sequences or the genome sequences from where the RdRp sequence obtained are as follows: Q9J5U7.1, BAU68080.1, NP_077730.1, YP_009160324.1, YP_009159826.1, YP_009143521.1, YP_009111338.1, YP_009032634.1, YP_009026384.1, YP_007761581.1, NP_042510.2, YP_003622540.1, YP_001718499.1, YP_001040002.1, NP_853560.2, YP_145791.1, YP_052864.1, NP_945134.1, NP_932306.1, NP_919040.1, NP_851403.1, NP_690806.1, NP_690839.1, NP_663297.1, NP_620648.1, NP_620564.1, NP_619751.1, NP_613283.1, NP_604464.1, NP_599247.1, NP_056825.2, NP_203553.1, NP_116487.1, NP_068549.1, NP_066241.1, NP_037647.1, NP_049374.1, NP_044335.1, NP_041277.1, ASU47554.1, ANH71250.1, CAB95006.3, AFR34027.1, ACU32794.1, AEH26189.1, AEH26193.1, QAY29244.1, YP_009552019.1, YP_009337113.1, QED21532.1, ANG56339.1, YP_009336934.1, AZS32325.1, YP_009333257.1, YP_009345113.1, YP_009337165.1.

### 2.15. Statistical Modeling of AnBV-1 Prevalence in the Field

To explain the prevalence of AnBV-1 in honey bees, we used three explanatory variables: (1) infected *Andrena* forager density, (2) the species diversity of floral resources utilized by honey bees in the habitat, and (3) the abundance of floral species that were shared between *Andrena* and honey bees in each site (scaled to 10,000 floral units to enable model convergence). Infected *Andrena* forager density was estimated for each site using the proportion of virus-positive *Andrena* individuals from the virological sample (median *n* = 11 per site), multiplied by the number of *Andrena* individuals collected in the bee forager sampling. The diversity of floral resources utilized by honey bees in each site was calculated using the Shannon diversity index, weighing species abundances based on their relative utilization by honey bees, as observed across all sites in the survey.

We used package lme4 in R version 4.0.3 to fit the data with generalized linear mixed effect models (GLMMs) with binomially-distributed errors and logit link functions [110,121]. The infection status (AnBV-1 present/absent) of individual honey bees was used as the response variable. The saturated model included all three explanatory variables, and the interaction term between infected *Andrena* density and shared flower abundance. Sites were included in all models as a random intercept factor. We also fitted reduced forms of the saturated model, as well as a null model that included only the site random factor. All models were ranked by their AICc scores using package bbmle [122]. The model with the lowest AICc score was selected as the best model and used for inference, and any model with ΔAICc >2 was considered significantly weaker. The overall significance of the best model is represented by its ΔAICc from the null. Model diagnostics were performed using package DHARM [123]. To exclude the possibility of spatial autocorrelation between adjacent plots (direct statistical testing is not possible in a sample of 14 sites [124]), we examined the distances between the three sites (Ramat Beit Shemesh 2, Luzit, and Srigim) that had the greatest AnBV-1 prevalence in honey bees, and found that they were far apart from each other (11.5 km, 7.9 km, and 4.4 km) and beyond the typical ~2 km foraging range of the majority of honey bee foragers) [22], whereas their five nearest sites (within 2 km) had zero AnBV-1 prevalence in honey bees.

## 3. Results and Discussion

### 3.1. Metatranscriptome Assessment of Viruses in Sympatric Bee Species

To investigate the degree of virus overlap and transmission between sympatric bee species in different environments, we collected samples of foraging bees from 14 sites in Israel. These sites were classified as high or low floral diversity and high or low honey bee activity and sampled on a single date between 13 March and 1 April 2018 (Figure 1). Over 2323 bees were observed and 1331 were collected. The most predominant bee taxa were *Apis mellifera* (honey bees, *n* = 263) and *Andrena* species, commonly known as mining bees (*n* = 876). Therefore, these bee taxa were the focus of our investigation (Figure 1 and Appendix A). The *Andrena* species in this study included *A. combusta ocraceohirta*, *A. aerinifrons levantina*, and *A. urfanella*, and an unidentified morphospecies of the subgenus *Truncandrena*. Distinct bee species may differentially contribute to interspecific virus spread. However, we considered these congeneric *Andrena* species as a single epidemiological population, because they are closely related phylogenetically and also share similar ecological niches with respect to phenology, dial activity, nesting habits, and diet. They are yellow mustard specialists, and we observed that greater than 96% of their floral visits were on *Sinapis alba* and *Hirschfeldia incana.*

The majority of honey bee-infecting viruses are single-stranded positive-sense RNA viruses (+ssRNA) (reviewed in [42,43,44]). To identify bee-associated viruses, RNA was extracted from 9 to 11 individual bees per taxon per site, and samples were pooled by bee taxon (i.e., *Apis mellifera* or *Andrena spp*.) for each of eight sites representing four distinct classifications including: low floral diversity and low honey bee activity, high floral diversity and high honey bee activity, high floral diversity and low honey bee activity, and low floral diversity and high honey bee activity. Ten species-specific sequencing libraries were generated from honey bees and mining bees obtained from eight representative sites (Figure 1). In addition, two virus-augmented libraries (one from *Apis mellifera* samples and one from *Andrena spp*. samples) were prepared and sequenced on an Illumina Hiseq (paired-end 200 nt) resulting in an average of 44.4 million reads per bee transcriptome library and 8 million reads per virus-augmented library (Appendix A, BioProject ID PRJNA687318 SRA files). The majority of reads in the *A. mellifera* sequencing libraries aligned to the *A. mellifera* genome (Amel_HAv3.1) and were removed from further analysis, as were any reads in the *Andrena spp.* libraries that aligned to the *A. mellifera* genome (Appendix A) [125,126]. The remaining reads (i.e., non-honey bee) were then aligned to previously characterized honey bee virus and virus-like sequences [49,59,60,100,104,127,128,129,130] (Appendix A). The most frequently detected viruses in these sequencing libraries included black queen cell virus (BQCV), deformed wing virus (DWV), sacbrood virus (SBV), the DNA virus Apis mellifera filamentous virus (AmFv), and members of the more sequence-divergent Lake Sinai virus group [111,131,132,133]. To facilitate discovery of previously unreported virus strains and virus-like sequences, sequence reads corresponding to non-viral pathogens, including the parasites *Varroa destructor* and *Lotmaria passim* (formerly known as *Crithidia mellificae*, strain sf), were removed [100,104,130]. The remaining sequences were de novo assembled into contiguous overlapping DNA segments (contigs) using Trinity [134]. The most abundant contigs were further characterized and described herein as a novel, bipartite, positive-sense single-stranded RNA virus, Andrena-associated bee virus-1 (AnBV-1, MW397640).

### 3.2. Sequence Variants of Bee-Infecting Viruses: Black Queen Cell Virus, Deformed Wing Virus, and Members of the Lake Sinai Virus Group 

RNA viruses replicate using virally encoded RNA-dependent RNA polymerases (RdRp), which typically have higher error rates than DNA-dependent DNA polymerases and often lack proof-reading capabilities [135]. Therefore, an actively replicating RNA virus results in millions/billions of sequence variants, referred to as quasispecies [136,137,138]. The degree of variation that warrants a new virus strain is not well established for all viruses and thus the term “sequence variant” is often used to describe a sequence that differs from previously reported viral genomes.

To assess variation in this sample cohort, sequences corresponding to well-characterized honey bee-infecting viruses, including BQCV, DWV-A, DWV-B/VDV-1, SBV, and representative LSVs were binned following identification using BLAST and assembled into contigs [106]. Unique representative contigs were aligned to corresponding viral genome reference sequences in order to identify the most similar viral genome currently reported in the NCBI database, which was used as a scaffold for contig assembly. Consensus sequences that represent the majority sequences of the viral variants described herein (i.e., Israel-2018 variants) were deposited in the NCBI database.

This analysis revealed nucleotide differences in black queen cell virus, deformed wing virus, and Lake Sinai virus genomes that may have precluded reliable detection of these viruses using commonly employed primers, which were developed based on virus genome sequences obtained from other geographic locations (e.g., United States, Germany, Belgium, and China). Therefore, new qPCR primers were designed for future quantification of the relative virus abundance in honey bee and mining bee samples obtained in this study (Appendix A).

#### 3.2.1. Black queen cell virus

The BQCV variant identified in this study, BQCV Israel-2018 (MW397638), shares the greatest percent nucleotide identity (i.e., 90%) with a BQCV variant (MH267693) associated with bees obtained in 2009 from mite-resistant honey bee colonies on the Island of Gotland, Sweden (Appendix A) [139]. Nucleotide alignment of the consensus sequence identified herein revealed numerous single-nucleotide polymorphisms, which were avoided in qPCR primer design. The depth sequence coverage for BQCV Israel-2018 varied by site and was greatest in the *Apis mellifera* library generated from the Ramat Beit Shemesh (RBSh) site (i.e., up to 6601x coverage) (Figure 2). This sampling site was dominated by *Hirschfeldia incana* (Mediterranean mustard) and a high degree of honey bee activity (Figure 1 and Appendix A).

Deformed wing virus strains commonly infect honey bees, replicate in and are transmitted by *Varroa destructor* mites that parasitize honey bees, and infect native and wild bee species and other insects [52,55,65,79,80,86,88,140,141]. Infection of developing honey bees may result in wing deformities that result in death during or shortly after emergence, whereas viral load and the ill effects of virus infection vary in adults, which can harbor between 1 × 10^5^ to ≥ 5 × 10^12^ viral RNA copies per bee (estimated from qPCR values ≥2 × 10^9^ copes/100 ng RNA and a total RNA per bee ~50 mg) [61,89,91,142,143,144]. Deformed wing virus infections have been associated with small colony population size and overwinter losses [89,95,145], and DWV infections coupled with high infestation of *Varroa destructor* mites, which also vector DWV, can kill colonies [69,78,146,147,148].

The DWV strains described in the literature include DWV-A, DWV-B (also called VDV-1), and DWV-C, as well as recombinant viruses [127,144,149,150]. DWV-A and DWV-B share approximately 84% nucleotide identity [70], and DWV-A/B recombinants are the most predominantly reported in literature in the US, Israel, and Europe (reviewed in [42,45,146]). Experimental evidence indicates that pupae infected with either DWV-A or DWV-B, and recombinant DWV-A/B viruses, can develop deformed wing pathology [144,151,152]. The deformed wing virus consensus sequence obtained in this study, DWV Israel-2018 (MW397639), is 97.3% identical to a recombinant DWV sequence (HM067438), which was originally described as having 5′ and 3’ untranslated regions (UTRs) that are more similar to DWV-A, and a central protein encoding region more similar to DWV-B [127,144] (Appendix A). Although DWV Israel-2018 and DWV-HM067438 share 97.3% nucleotide identity across most of the genome, they are divergent in a small region within the 5′ end of the polyprotein coding sequence (i.e., nucleotides 1160–1540). In this small region, DWV Israel-2018 best aligns to DWV-A (NC_004830; Appendix A). DWV Israel-2018 is similar to DWV-B strains in that it has an 11 nt gap when aligned with DWV-A strains (Appendix A). Together, this and other studies indicate that recombinant strains are prevalent and abundant. Overall, the extent of DWV variability and abundance may be under appreciated since most studies quantify viruses using primers that amplify short products from conserved regions of the genome, including the RNA-dependent RNA polymerase encoding region (Figure 2). The depth of sequence coverage was highest in the *Apis mellifera* library generated from the Galon3 site (GB3) which was dominated by white mustard (*Sinapis alba*), but had low honey bee activity on the sample date (Figure 2). The depth of sequence coverage was very high over most of the genome (i.e., up to 121,593× coverage).

#### 3.2.2. Lake Sinai virus

Lake Sinai virus-1 (LSV-1) and Lake Sinai virus-2 (LSV-2) were the first two viruses identified in what is now recognized as the *Sinaivirus* group, which includes over 200 NCBI sequence entries ranging in nucleotide identity from ~63% to 100% (Appendix A) [61,111,132]. LSV-1 and LSV-2, which serve as type species for this group, are 71% identical at the nucleotide level and are also distinguished by differences in their protein coding regions [61]. The LSV-1 capsid encoding region overlaps with the RdRp gene in the +1 reading frame for 125 nt before ending in a pair of stop codons, whereas the LSV-2 capsid is in frame with the RdRp [61]. The protein coding regions of LSV-2 are separated by 18 nt, whereas other isolates, including those described from a 2015 Belgium sample cohort, contain a variable spacer (19–23 nt) between the capsid protein and the RdRp protein encoding region [153].

The majority of the unique LSV contigs in this study were most similar to LSV-2 (HQ888865). The LSV-2 contigs included the longest (up to 5997 nt) and most abundant contig (2.4 million estimate counts or 49,514 transcripts per million (tpm)), which accounted for 37.4% of the LSV-2 contigs that were clearly assigned to a particular NCBI designated LSV species/strain. Contigs most similar to LSV-NE (NC_035113) were also well represented in sequence libraries (i.e., accounting for 41% of easily assigned unique contigs, including a 5952 nt in length contig with an estimated read count of 251,659).

The LSV-2 Israel-2018 consensus sequence (MW397637) identified in this study, via alignment and assembly of unique LSV-2 contigs >400 nt, was 77% identical to LSV-2 (HQ888865) (Appendix A). Interestingly, the LSV-2 Israel-2018 consensus sequence and LSV-2 HQ888865 sequence share less nucleotide identity from nucleotide position 2006 to 2156, whereas this region of LSV-1 (HQ871931) and LSV-2 (HQ888865) virus strains is identical (i.e., 2105−2231). LSV-2 Israel-2018 sequence coverage was greatest in the *Apis mellifera* library from the RBSh site, with up to 326,028x coverage in the middle of the genome and over 50,000x coverage at the 5′ and 3′ ends of the genome (Figure 2). The capsid and RdRp encoding regions of this variant were separated by 19 nucleotides.

The LSV-NE Israel-2018 consensus sequence (MW397636) is most similar to LSV-NE sequence (NC_035113) with which it shares 84% nucleotide identity (Appendix A). Unique contigs (>400 nt) assembled from RNAseq reads provided up to 12× contig coverage. However, it is important to note that this does not depict true sequencing depth, which is illustrated in Figure 2 with coverage of up to 198,843× in the *Apis mellifera* sequencing library from the Berko2 (B2) site, which had high floral diversity and low honey bee activity. LSV-NE contig 51 was the longest (5952 nt), well-represented contig (i.e., 9209 tpm and 251,659 estimated read counts (Appendix A)).

High-throughput sequencing of representative samples obtained from this sample cohort was essential in the development of new qPCR primers designed to detect the viral variants in this study (Appendix A). This result underscores the importance of full sequence characterization prior to the assessment of relative viral abundance in samples obtained in different parts of the world, and likely for any sample cohort due to the quasispecies nature of RNA viruses and the potential for consensus sequence drift over time in all geographic areas. High-throughput sequencing was used to characterize the virus genomes in this study, but sequencing across a larger target nucleotide range using multiple primer sets would likely be sufficient. In addition, for some studies, designing primers for more conserved areas of the genome (i.e., the RdRp encoding region) may be best, whereas other studies may require specific primer sets for particular viral strains [111,149].

### 3.3. Novel Bipartite Positive-Sense RNA Virus Identified in Andrena Mining Bees

A novel bipartite positive-sense single-stranded RNA virus, Andrena-associated bee virus-1 (AnBV-1), was discovered via high-throughput sequencing of RNA samples from mining bees. Virus discovery efforts focused on sequence data obtained from “virus-augmented” sequence libraries, which were generated from RNA isolated from nuclease treated lysates of pooled mining bee or honey bee samples. The purpose of nuclease treatment prior to RNA isolation was to digest all unprotected, non-encapsidated nucleic acid in bee lysates and thus augment the amount of nucleic acid that was protected by virus capsids in sequencing libraries. Assembly of contigs, using Trinity, from reads that did not align with the *Apis mellifera* genome nor with the non-viral sequences in the bee holobiome revealed two large, novel contigs that we named AnBV-1 RNA 1 (2005 nt) and RNA 2 (2721 nt) (Appendix A) [100,134]. These contigs were the most abundant virus-like contigs in the *Andrena spp.* virus-augmented library. They are also unique, as nucleotide BLAST analyses of either RNA segment resulted in no significant returns. To ensure that these contigs were not the result of assembly errors, the sequences were confirmed by PCR amplification coupled with Sanger (chain termination) sequencing (Appendix A). Attempts to bridge the two contigs using combinations of forward and reverse primers from the individual contigs failed, and thus supported assembly data indicating that the AnBV-1 genome is bipartite (Appendix A). Collectively, these data suggest that the two contigs are derived from AnBV-1, a novel bipartite positive-sense single-stranded RNA virus (MW397640, MW397641).

High-throughput sequencing coverage of AnBV-1 was greatest in the library prepared from *Andrena spp.* collected at the Ramat Beit Shemesh (RBSh) site, which had low floral diversity and high honey bee activity. AnBV-1 coverage reached up to 1.45 × 10^6^-fold for RNA 1 and up to 5.9 × 10^5^-fold for RNA 2 (Figure 2). AnBV-1 coverage was also high from RNA extracted from a virus-augmented (i.e., nuclease treated) lysate obtained from a single *Andrena* bee sample that was positive for AnBV-1, but not positive for several other bee viruses, as assessed by virus/sequence-specific polymerase chain reaction (PCR) (i.e., LSV-U, IAPV, CBPV, KBV, and ABPV) and quantitative PCR for SBV, BQCV, LSV, DWV, and AmFV (Appendix A). Rapid Amplification of cDNA Ends (RACE) in conjunction with Sanger sequencing was performed to confirm the sequence of RNA termini, which notoriously difficult to sequence due to potential RNA structure and/or repetitive sequence. 

AnBV-1 open reading frames (ORFs) were predicted in SnapGene Viewer and Geneious Prime (Figure 3A and Appendix A). Predicted ORF sequences and their translations were queried with BLAST and PHMMER online utilities [113,114]. AnBV-1 RNA 1 encodes four predicted ORFs. The largest, ORF1, is 1740 nt in length and encodes a putative 540 aa protein. Initial BLAST analyses did not identify orthologous proteins and/or motifs in ORF1, ORF3, or ORF4. However, PHMMER analyses indicated that ORF2 of RNA 1 encodes a 161 amino acid (aa) virion protein based on its similarity to a putative virion protein from spider-associated Loderio virus (NC_031748) [115]. AnBV-1 RNA 1 ORF2 shares 25.7% aa identity with Loderio virus ORF3, which is a 193 aa protein in the European Molecular Biology Laboratory database, and both of these proteins fall into the SP24 family (PF16504, Uniprot A0A1D9CFP5). This protein family includes SP24, which is a 24 kD structural protein, in a family of putative virion membrane proteins of plant and insect viruses, including the honey bee-infecting chronic bee paralysis virus. Chronic bee paralysis virus (CBPV), like AnBV-1, has a bipartite genome structure [154]. The putative CBPV capsid protein is a 183 aa protein encoded by ORF3 of RNA 2 (NC_010712). The putative AnBV-1 and CBPV capsid proteins share 21.3% aa identity. Further support of AnBV-1 RNA 1 ORF2 encoding a virion protein was obtained from an HHpred search of the Pfam database (Pfam-A v33.1) which returned the ‘SP24; putative virion membrane protein of plant and insect virus’ as a top match (e-value = 0.00091) [155].

Analysis of the two largest open reading frames of AnBV-1 RNA 2 using BLASTp indicates that ORF2 encodes the putative RNA-dependent RNA polymerase (RdRp) (Appendix A). The putative 499 aa AnBV-1 RdRp sequence produces the strongest alignment with a putative Castleton Burn virus RNA-dependent RNA polymerase, (BLASTp e-value = 0), which was identified by sequencing bumble bee samples (Appendix A) [156]. The RNA-dependent RNA polymerase proteins of AnBV-1 and Castleton Burn virus are similar. However, they share only 53.3% amino acid identity (Appendix A).

To gain further insight into the relationship between AnBV-1 and other insect-infecting viruses, including common well-characterized honey bee viruses in the *Dicistroviridae* (i.e., ABPV, BQCV, IAPV, KBV), *Iflaviridae* (i.e., DWV-A, DWV-B/VDV-1), and *Sinaiviridae* (e.g., LSV-1, LSV-2, LSV-NE) families, we constructed a phylogenetic tree of viral RdRp amino acid sequences. This analysis revealed that the AnBV-1 RdRp sequence forms a well-supported monophyletic clade with sequences from tombus-like viruses. In particular, these results indicate that, based on their RdRp sequences, AnBV-1 shares a most recent common ancestor with Castleton Burn virus (Figure 3).

Sequence and phylogenetic data indicate that AnBV-1 is a positive-sense single-stranded RNA virus (+ssRNA). Therefore, like other +ssRNA viruses, a negative-strand copy of the viral genome must be produced during viral replication, as it serves as the template for the production of viral transcripts and full-length genome segments. Strand-specific reverse-transcription (RT)-PCR was performed on RNA isolated from AnBV-1-positive samples. Visualization of AnBV-1 RNA 1 and RNA 2 products from negative-strand targeting reactions indicate that AnBV-1 productively infects bees (Appendix A).

### 3.4. Andrena-associated bee virus-1 (AnBV-1) is More Prevalent in Mining Bees

The prevalence and abundance of AnBV-1 in honey bees and mining bees, which represents a single sampling date at each of 14 sites, was assessed at the individual bee level by qPCR (Appendix A). AnBV-1 was more prevalent in mining bees. Specifically, 71 of the 148 mining bee samples analyzed were AnBV-1 positive (48%), whereas 22 of the 143 honey bee samples analyzed were AnBV-1 positive (15%) (X^2^ = 44.057, df = 1, and * *p*-value = 3 × 10^−11^) (Figure 4). Comparison of AnBV-1 RNA 1 or RNA 2 copy number in individual AnBV-1-positive bee samples indicate that AnBV-1 RNA 1 abundance was greater in mining bees compared to honey bee samples (*p* = 0.0279), whereas the difference in RNA 2 abundance was not significant, *p* = 0.29 (Figure 4). Since qPCR values represent differences in RNA abundance, including both genomic and messenger RNAs (transcripts), differences in the quantities of the AnBV-1 RNA segments may be biological and/or due to differences in qPCR efficiencies (Appendix A). Together, these data indicate that AnBV-1 is more prevalent and abundant in mining bees.

### 3.5. AnBV-1 Replicates in Primary Honey Bee Pupal Cell Cultures

Since AnBV-1 was detected in honey bees, albeit in lower frequency and abundance, and honey bee cells support the replication of other bee-infecting viruses including SBV, IAPV, DWV, and BQCV [157,158,159], we examined the ability of AnBV-1 to infect and replicate in primary honey bee pupal cells maintained in culture. The inoculum for infection studies was clarified lysate obtained from a single *Andrena* sample with high AnBV-1 abundance (1.3–2.6 × 10^8^ copies AnBV-1 per µl lysate). Mock infections were carried out with clarified lysate obtained from a separate individual *Andrena* sample without detectable levels of AnBV-1 (Figure 5). Samples from three independent replicates of honey bee pupal cell cultures were harvested at 0, 1, 2, 4, 5, and 7 days post-infection (dpi) (Figure 5 and Appendix A). Assessment of the abundance of AnBV-1 RNA 1 and RNA 2 by qPCR indicated virus replication by 4 or 5 days post-infection in three replicates (Appendix A). The abundance of housekeeping gene Rpl8 was consistent in each AnBV-1 and mock sample in all replicates, indicating that the cells were not dying, senescing, nor differentially replicating during the course of this experiment (Appendix A). When combined, median AnBV-1 RNA 1 levels at 4 dpi (3.6 × 10^7^ +/−2.9 × 10^6^ copies per 24 ng RNA) were greater than at 0 dpi (1.4 × 10^7^ +/−1.5 × 10^6^ copies per 24 ng RNA) (Figure 5A, left panel, *p*-value = 0.0714). Though a similar trend was observed for AnBV-1 RNA 2, the increase was not statistically significant until 7 dpi (5 × 10^7^ +/−2.5 × 10^6^ copies per 24 ng RNA) as compared to 0 dpi (1.2 × 10^7^ +/−3.4 × 10^6^ copies per 24 ng RNA) (Figure 5A, right panel, *p*-value = 0.0095), which is likely a reflection of the low sample size (i.e., *n* = 2−6 per time point) due to the limited availability of honey bee pupal cells for infection in the late fall in Montana, when honey bee queens cease egg laying in preparation for winter. Together the increasing trend in AnBV-1 RNA, which persisted until the conclusion of the experiments at 7 dpi, indicates that AnBV-1 replicates in honey bee pupal cells (Figure 5 and Appendix A). To provide further evidence of AnBV-1 replication in cultured primary honey bee cells, strand-specific qPCR was carried out on select samples. Results indicate an upward trend in AnBV-1-negative strand abundance over the course of infection (Figure 5B). Levels of AnBV-1 RNA 1 and RNA 2 were approximately 2–3 fold greater at 7 dpi as compared to 1 dpi, confirming that AnBV-1 replicates in primary honey bee pupal cells.

### 3.6. Sequence Analyses Indicate Virus Transmission between Sympatric Bee Species

Previous studies that indicated viruses were shared among sympatric bee taxa based their conclusions on the phylogenetic similarity of viral RdRp or capsid amino acid sequences, which represent only a portion of the viral genomes [54,56,64,87,160]. Herein, we compared nucleotide identity over entire viral genomes. Viral genome sequences with greater than 90% nucleotide identity are likely shared, as this level of variation can be easily explained by the RNA diversity generated during viral genomic replication [137]. Virus consensus sequence data from specific sites and from the virus-augmented sequencing libraries indicate that viruses are transmitted between honey bees and mining bees. However, sequence data from samples pooled by site and by bee taxa indicate the suite of viruses in sympatric honey bees and mining bees was not always shared. Sequence data indicate that AnBV-1 is transmitted between mining bees and honey bees, as the consensus sequences obtained from both site-specific and virus-augmented sequencing libraries share over 99% nucleotide identity. Likewise, DWV virus transmission between species was supported by 92% nt sequence identity in the RbSh site and in the entire sample cohort (i.e., virus-augmented libraries shared 97% nt identity). Similarly, interspecies transmission of BQCV sequences was indicated at the RBSh site (i.e., 96% nt identity), though the coverage was lower in the mining bee sample compared to the honey bee sample. The high degree of similarity in the Lake Sinai virus group made it difficult to assess potential transmission between bee species. Sequence data indicated that LSV-NE is transmitted between honey bees and mining bees, which shared over 97% nucleotide identity in the regions of the genome with sufficient coverage in the mining bee sample (i.e., ~50% of the genome, nt positions 700–1600 and 2800 to 4400), whereas the potential of LSV-2 transmission could not be adequately assessed by sequence analysis alone. Together, data from this single-time point, multiple-site observational cohort study indicate that viruses are transmitted between honey bees and mining bees. Sequence analysis alone is insufficient to assess the directionality of transmission. Furthermore, it is likely that for many bee-infecting viruses, interspecific virus transmission is multidirectional (i.e., viruses are transmitted among several bee and other insect species), and influenced by additional parameters including parasitic vectors, co-infection status, species density, habitat quality, and sample date. Therefore, studies that longitudinally monitor the prevalence and abundance in the context of numerous other factors that may impact virus transmission are required in order to better understand bee virus ecology.

### 3.7. The Probability of AnBV-1 Infection in Honey Bees Is Modulated by the Floral Community

Since interspecies virus transmission may occur between sympatric bee species foraging on shared floral resources, we investigated patterns of virus spread in honey bee and mining bee populations in the context of the floral community and infected mining bee density in the habitat. The field data were fit with statistical models of AnBV-1 prevalence in honey bees, with the following explanatory variables: (i) the density of AnBV-1-infected mining bees, (ii) the species diversity of flowers utilized by honey bees, and (iii) the abundance of flowers of plant species visited by both mining bees and honey bees. In this study, honey bees, which are generalist pollinators, visited more flower species than the mining bees, which preferred to forage on yellow mustards (i.e., Brassicaceous plants) (Appendix A). In the best fit model, AnBV-1 infection prevalence in honey bees was most strongly associated with low floral diversity (*p* < 0.0001) (Figure 6B). Indeed, AnBV-1 infections in honey bees were detected in sites with low floral diversity and were absent in honey bees in high floral diversity sites (Figure 6C and Appendix A). These results suggest a strong effect of epidemiological dilution by flower species diversity [161,162,163,164]; in sites with high floral diversity, the generalist honey bees can forage on a wide range of floral species, and therefore their exposure to flowers that are visited by mining bees, which primarily forage on yellow mustard plants, is reduced. The correlation between low flower diversity and AnBV-1 infection in honey bees can also be explained by other, non-mutually-exclusive mechanisms. For example, honey bees foraging in sites with high floral diversity may benefit from higher quality nutritional resources, which may increase their resilience against virus infections [141,165,166,167].

The density of AnBV-1-infected mining bees alone did not have a consistent effect on AnBV-1 infection prevalence in honey bees (*p* = 0.1496) (Figure 6B). However, its effect varied in magnitude, depending on the abundance of shared floral species between mining bees and honey bees (interaction *p* = 0.0038) (Figure 6C). This suggests, epidemiological dilution by flower abundance. Since flowers can serve as virus transmission hubs [161], the number of AnBV-1 contaminated flowers can be diluted by many non-contaminated flowers, reducing the probability of sharing the same individual flowers between an infected mining bee and an uninfected honey bee. At very high densities of AnBV-1-infected mining bees, the impact of shared flower abundance in sites with low floral diversity appears to reverse (Figure 6B, red lines), perhaps because of honey bee recruitment to dominant floral resources [168]. In habitats with low floral diversity and a low abundance of flowers (e.g., Luzit) that are shared with mining bees, AnBV-1 infection in honey bees can occur even in the absence of detected AnBV-1 infection in mining bees (Figure 6B). The observation that AnBV-1 infections were detected only in honey bees at the Luzit site, which had low floral diversity and high honey bee activity, indicates that AnBV-1 may also spread between co-foraging honey bees (Appendix A). Whereas the relationships observed here are consistent with those expected under interspecific transmission through shared flowers, and similar to reported results from other studies that followed the temporal dynamics of different pollinator pathogen infections [161], our field data were obtained on a single sample date, and it is likely that AnBV-1 infection prevalence in honey bees, mining bees, and other bees and insects not screened in our study varies with time. Therefore, prevalence at a particular sampling date represents the status of the system on a particular sampling date (or point in time).

## 4. Conclusions

In summary, data from this sample cohort underscore the importance of a broad investigation of the bee-associated virome and provide a framework for future investigation of bee virus ecology. The majority of bee-infecting viruses are RNA viruses, which often exist as quasispecies. Therefore, it is important to incorporate sequencing (e.g., high-throughput sequencing and/or targeted PCR with multiple primer sets per virus coupled with Sanger sequencing) to identify novel viruses and viral variants, prior to assessing viral prevalence and abundance at the individual bee, colony, and community levels. Virus discovery efforts have primarily focused on honey bees, but it is clear from this and other studies that the diversity of bee-infecting viruses is greater than currently appreciated. Likewise, detection of bee-infecting viruses, which are typically referred to as “honey bee-infecting viruses”, in a broad range of hymenopteran insects, indicates that investigation of interspecific virus transmission should consider a broader potential host range.

For this study, we opted to sample multiple sites, with distinct floral and bee communities, within a single blooming season and focus our analyses on the two most prevalent co-foraging bee taxa (i.e., *Apis mellifera* and four specialist *Andrena spp.*) and a novel bipartite +ssRNA virus named Andrena-associated bee virus-1 (AnBV-1) that replicates in mining bees, honey bees, and primary honey bee pupal cell culture. AnBV-1 was most prevalent and abundant in mining bees, though it was also detected in honey bees, albeit at lower frequency. Statistical modeling that examined the roles of ecological factors, including floral diversity and infected mining bee density, indicated that the probability of AnBV-1 infection in honey bees is greater in low-floral-diversity habitats, and suggested that interspecific transmission is strongly modulated by the floral community in the habitat. Future studies that examine the temporal dynamics of bee/insect density, virus prevalence and abundance, and habitat-associated parameters will be important to reveal patterns of virus transmission, clearance, and reinfection of bee taxa within bee communities. Likewise, studies aimed at understanding the impact of AnBV-1 on individual bees, bee colonies, and bee communities are required. It will also be interesting to investigate the impact of multiple viruses (individually and in the context of other co-infections) from the individual bee to community levels to determine their impact on bee behavior and pollination activity, and/or survival. The data from this study suggest effects of epidemiological dilution by floral diversity and abundance, and that interspecific virus transmission is modulated by the floral community in the habitat. Therefore, land management strategies that aim to enhance floral diversity and abundance may reduce AnBV-1 (and possibly other viruses) spread between co-foraging bees. Regardless of the impact of the bee/insect community virome, efforts aimed at enhancing and/or maintaining bee forage in managed and wild landscapes will likely benefit all plant and animal communities.

## Figures and Tables

**Figure 1 viruses-13-00291-f001:**
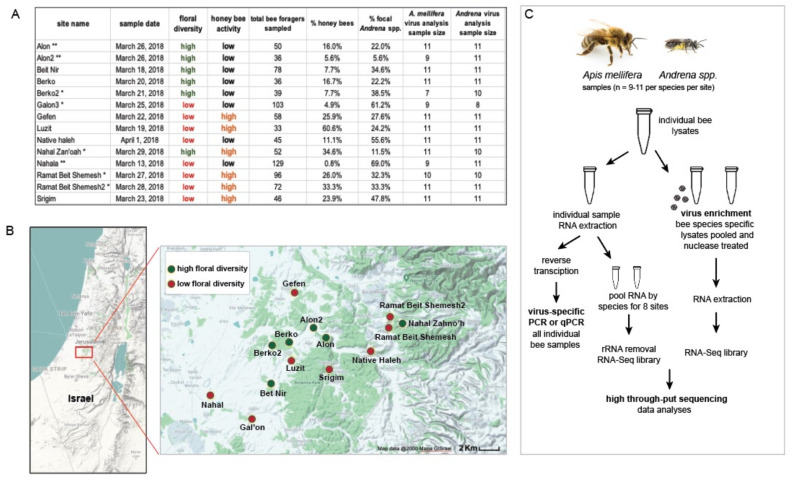
Overview of sampling and analysis of sympatric bee species collected from 14 sites in Israel. (**A**). The table provides a summary of the site-specific data obtained in this 2018 sample cohort. Sampling sites were selected for the survey as either high or low floral diversity, based on their floral species richness. Honey bee activity at each site was also categorized as either high or low, based on the percentage of honey bees out of the total number of bees sampled. This study resulted in the collection of over 1331 bee samples, including 263 *Apis mellifera* and 876 *Andrena spp.* samples. Assessment of viral diversity and abundance was carried out on a subset of individual bees (i.e., a median of 11 individual bees per taxon per site). (**B**). Map of Israel with sample sites labeled and marked with green and red dots indicating high and low floral diversity, respectively. (**C**). Schematic of the sample sizes and molecular analyses.

**Figure 2 viruses-13-00291-f002:**
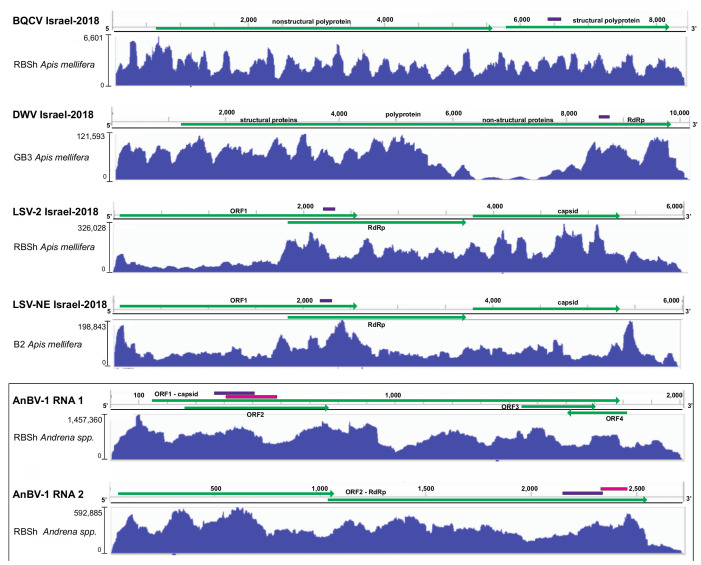
Sequence coverage of virus genome variants characterized by high-throughput sequencing and virus-specific qPCR. Reads in the sequence library with the highest number of reads were aligned to the consensus genome variants identified in this study including BQCV Israel-2018 (MW397638), DWV Israel-2018 (MW397639), LSV-2 Israel-2018 (MW397637), and LSV-NE Israel-2018 (MW397636), and to both genome segments of Andrena-associated bee virus-1 (AnBV-1), the novel bicistronic (+) ssRNA virus identified in this study (MW397640, MW397640). Vertical and horizontal Scheme 3.2.2. Deformed wing virus.

**Figure 3 viruses-13-00291-f003:**
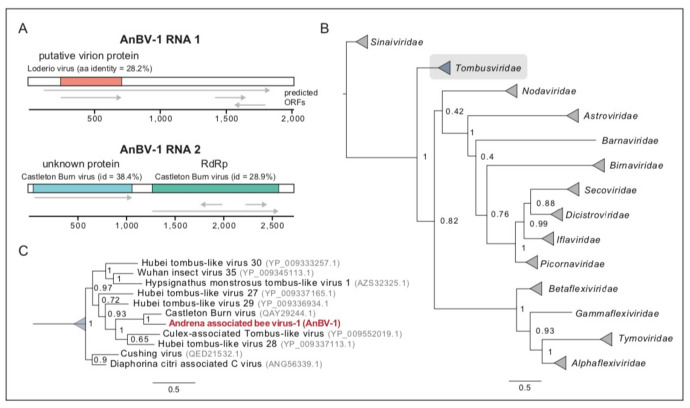
Andrena-associated bee virus-1 is a bipartite RNA tombus-like virus. (**A**). Schematic diagram of AnBV-1 genome, RNA 1 and RNA 2, with predicted open reading frames (ORFs) labeled with gene annotations based on PHMMER, HHpred and BLAST analyses and corresponding percent amino acid identity. (**B**). The AnBV-1 RdRp sequence forms a well-supported monophyletic clade with sequences in the *Tombusviridae* family (highlighted in gray). The AnBV-1 phylogenetic relationship was inferred from a maximum likelihood consensus tree of insect viruses, including previously described honey bee viruses in the *Sinaiviridae*, *Dicistroviridae*, and *Iflaviridae* families. Numbers on branches are Bayesian posterior probabilities (0–1); the scale bar indicates substitutions per site, and GenBank accession numbers for either the RdRp sequences or the genome sequences from where the RdRp sequence obtained are in parentheses. (**C**). Detailed view of the *Tombusviridae* clade in panel (**B**), illustrated that AnBV-1 RdRp is most similar to Castleton burn virus and Hubei tombus-like viruses. GenBank accessions are indicated in gray.

**Figure 4 viruses-13-00291-f004:**
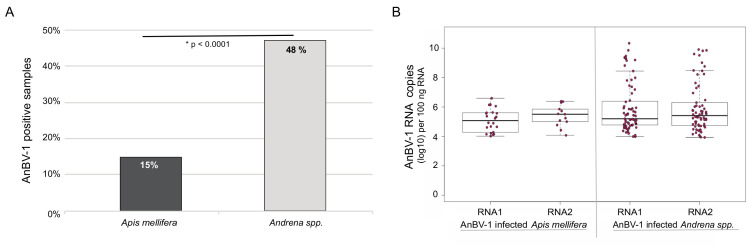
Andrena-associated bee virus-1 (AnBV-1) is more prevalent in *Andrena.* (**A**). AnBV-1 prevalence in this sample cohort, which represents a single sampling date at each of 14 sites, was determined from binary analysis of quantitative PCR (i.e., a sample was designated either positive (1) or negative (0) for AnBV-1) and presented as the percent AnBV-1 positive samples for each species. Specifically, 22 of the 143 *Apis mellifera* samples analyzed were AnBV-1 positive (15%) and 71 of the 148 *Andrena spp.* samples analyzed were AnBV-1 positive (48%). The data were analyzed using a chi square test for proportions, where X^2^ = 44.057, df = 1, and * *p*-value = 3 × 10^−11^. (**B**). Quantitative PCR was used to determine the AnBV-1 RNA 1 or RNA 2 copy number in individual AnBV-1-positive *Apis mellifera* and *Andrena spp*. samples. Samples with undetectable AnBV-1 levels were not graphed or analyzed. Comparison of the RNA abundance between species using Dunnett’s test indicates that AnBV-1 RNA 1 abundance was greater in *Andrena spp.* samples compared *to Apis mellifera* samples (*p* = 0.0279), but RNA 2 abundance between the two species was not significant, *p* = 0.29.

**Figure 5 viruses-13-00291-f005:**
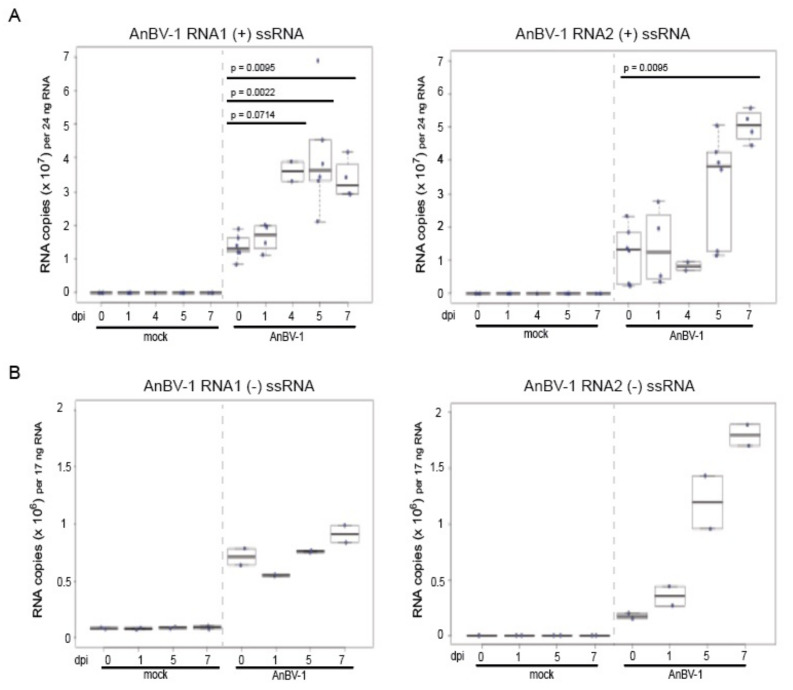
AnBV-1 replicates in honey bee pupal cells. To examine the ability of AnBV-1 to replicate in primary honey bee cells, cultures of pupal cells were incubated with AnBV-1-negative lysate (mock) or AnBV-1-positive lysate (AnBV-1). Total RNA was isolated from each cell culture well at 0, 1, 4, 5, or 7 days post-infection (dpi) and virus replication was assessed via strand-specific reverse transcription followed by qPCR. (**A**). Quantification of positive-strand, including both genome copies and transcripts, of AnBV-1 RNA 1 and AnBV-1 RNA 2 at each designated dpi. For RNA 1, 3.6 × 10^7^ +/−2.9 × 10^6^ copies per 24 ng RNA at 4 dpi vs. 1.4 × 10^7^ +/−1.5 × 10^6^ copies per 24 ng RNA at 0 dpi. For RNA 2, 5 × 10^7^ +/−2.5 × 10^6^ copies per 24 ng at 7 dpi vs. 1.2 × 10^7^ +/−3.4 × 10^6^ copies per 24 ng RNA at 0 dpi. (**B**). Quantitative assessment of negative-strand, replicative intermediate forms of AnBV-1 genome segments at each designated dpi; AnBV-1 RNA 1 and AnBV-1 RNA 2. Data were analyzed using R Studio and *p*-values were calculated using a pairwise Wilcoxon Rank Sums test with no multiple comparisons correction, and sample sizes per time point ranged from 2 to 6 samples. Relative quantity of the housekeeping gene *Amrpl8* was consistent across all samples at all time points.

**Figure 6 viruses-13-00291-f006:**
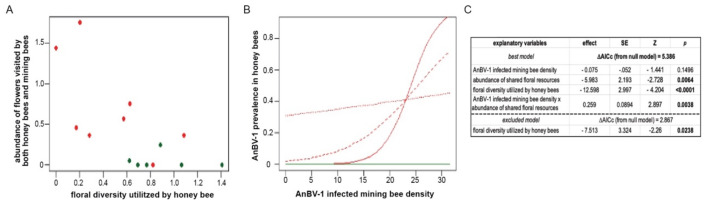
The probability of AnBV-1 infection in honey bees is greater in habitats with low floral diversity. (**A**). Patterns of bee resource use in high floral diversity sites (green) and low floral diversity sites (red), including the diversity of floral resources utilized by honey bees (horizontal axis) and the abundance of flower species that were shared between honey bees and mining bees (vertical axis); values scaled by 10,000^–1^ floral units. (**B**). Best statistical model for the roles of AnBV-1-infected mining bees and floral diversity on the probability of AnBV-1 infection in honey bees. All models are logistic generalized linear mixed models (GLMMs) for the probability of AnBV-1 presence in honey bees, with 143 observations. Scheme 2. The best model was contrasted with a null model containing only the random effect of site. (**C**). Best model fit to data, describing the relationship between AnBV-1 prevalence in honey bees (vertical axis) and three habitat characteristics: (i) the density of AnBV-1-infected mining bees, (ii) the diversity of floral resources utilized by honey bees (red curves = mean of low diversity sites, green curves = mean of high diversity sites), and (iii) the abundance of flower species that are shared between honey bees and mining bees (solid = maximal abundance observed in the habitat type, dotted = minimal abundance observed in the habitat type, dashed = mean abundance observed in the habitat type). For high-diversity habitats (in green), the three curves for different floral abundances overlap.

## Data Availability

The majority of the data generated or analyzed during this study are included in this published article and its Appendix A files (available online). Additional data are available from the corresponding author upon request, and sequence data are available on GenBank and the NCBI Sequence Read Archive (BioProject ID PRJNA687318). Custom bash and R scripts: https://github.com/charlieccarey/viral_transcriptome_of_bees.

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
