# Peer review of "Metatranscriptome Analysis of Sympatric Bee Species Identifies Bee Virus Variants and a New Virus, Andrena-Associated Bee Virus-1"

_viruses, 2021, doi:10.3390/v13020291_

Round 1
Reviewer 1 Report
This paper presents some points of strenght that are worth to be highlighted (apart from a rich, complete bibliography of the subject):
- the attention to virus genetics
- the community level approach in studying the epidemiology of bee viruses
- the laboratory based experiments with primary honey bee pupal cells maintained in culture that were carried out to confirm that 231 AnBV-1 replicates in honey bees, not to mention the negative-strand specific RT-PCR.
This makes the authors’ conclusions about AnBV-1 really interesting and novel; in particular, since they verified replication in honey bees cells.
Also sequence identity is important to support the interspecific transmission and it is nicely complemented by the ecological study showing how floral diversity can affect it. Under this respect, the lack of an experimental demonstration of interspecific transmission between Andrena and Apis, under controlled conditions, can be regarded as the only limit of this manuscript which, otherwise, would have been the perfect example of how similar studies should be carried out.
Thus my very positive evaluation of this manuscript.
Following are some minor points
Row 77
“Honey bees are eusocial insects that live in colonies of ~30,000 individuals”
If the colonies authors are speaking here are wild ones then this figure probably holds but if managed bee colonies are those of interest in this context (as the reasoning about colony losses would indicate) then 50000 individual is a more likely estimate.
Row 81
“Honey bees forage across longer distances (i.e., ~ 3-5 mile radius from their hive) compared to most other bee species”
Indeed worker bees can fly even further but on average they don’t forage farther than 1-2 km from their hive; therefore I’d specify whether that distance has to be intended as average, maximum, etc.
Row 104
“These three species” one would expect the name of the three species
Row 110
“fungi, microsporidia” as far as I know microsporidia are included into the Fungi taxa
Row 174
I don’t know if ref. 70 is appropriate here
Row 219
“within”?
Row 245
“which”?
Row 368
“more similar to DWV” -B?
Row 615
“Selected”?
Row 936
“ref.”?
Literature
Species names are not in intalic
Reviewer 2 Report
The manuscript describes the findings of a cross-sectional metatranscriptome-based virus screening in sympatric bee species, collected across 14 sites in Israel. They identified variants of four viruses as well as a novel one tentatively named as Andrena bee associated virus 1 (AnBV-1) which appears as a bipartite positive single strand RNA Tombo-like virus. They further assessed prevalence and abundance of the novel virus in mining and honey bees and revealed an elevated prevalence in habitats with low floral diversity via statistical modeling. They further designed new primers for amplification and examined virus replication in primary honey bee cells.
My only concern about the manuscript is that the text is too long, sometimes becoming hard to follow (1150 lines without abstract and acknowledgements). There are also too many supplements and references. However, it is consistent throughout, well-written and informative, which can be regarded as the authors’ style.
This reviewer has no further comments than the one below.
(line 304, 750 ) delete space (...viral species...) (...viruses is greater...)
Author Response
Please see the attachment. (one response letter for all Reviewers)

Reviewer 3 Report
This manuscript reports the investigation of viral diversity and host spectrum in sympatric bee species in Israel. The major findings of this paper are (i) the description of a new virus, primarily infecting mining bees, (ii) the presence of this virus in honeybees in certain sites where bees were sampled, (iii) the demonstration that this new virus AnBV1 is replicating in honeybees, and (iv) the correlation between AnBV1 prevalence in honeybees and local floral diversity.
I really enjoyed reading this ms, which is well written, and address many aspects of epidemiology, i.e. the discovery of a novel pathogen, the demonstration of replication, and an investigation of the ecological and environmental factors driving interspecies transmission. I particularly enjoyed the introduction, that was very well documented. Overall, this interdisciplinary study has been well thought, and the report is clear, with very interesting findings.
Most of the results are, to me, well analyzed, with a careful interpretation, and even clear attempts to address potential alternative hypothesis. In that sense, I have not much to say about the virus discovery aspect, the sequence analysis and the experimental infection, which were largely performed and reported in a very professional manner -but see minor comments below.
I have more concerns, however, about the analysis of disease transmission presented in this paper. In this ms, authors showed that AnBV1 prevalence in honeybees is higher in sites with low flower diversity. Their interpretation is that honeybees, in low flower diversity habitats, foraged on the same resources than mining bees, the primary host of AnBV1, and consequently became contaminated following visitation of contaminated flowers. In contrast, in more flower diverse habitats, honeybees may have foraged on other flower species, and thus reduce their chance of getting infected. I find this interpretation a bit too speculative, as the relationship between flower diversity and AnBV1 prevalence in honeybee could be explained by other mechanisms. I feel that data is missing to clearly conclude that flower diversity is key in the transmission of AnBV1. To me, one essential, but yet lacking, piece of information is the report of bee-flower visitation per site. Without reporting which flower species/site were visited by mining bees and honeybees, it is impossible to say if these two taxa were foraging on the same plant species. Analyzing what is called the niche overlap for each taxa and each site is crucial for the demonstration of transmission. Overall, we lack information on the flower composition of the sites, and the foraging spectrum of the bee taxa at each site.
In their conclusion (lines 764 & 774), authors claimed that floral community composition strongly modulated interspecific transmission of AnBV1. I totally disagree with this statement. I understand that flower diversity, and not the plant community composition (i.e. a matrix of abundance from each flower species) was used here. This detail matters, as the present study do not provide the evidence that on site where the virus was shared by honeybees and Andrena, both taxa were foraging on the same flower. They only showed that honeybees were foraging on less flower species, and thus potentially on the same flowers than mining bees. Therefore, claiming that plant diversity (and not community) is influencing transmission is still too speculative with the data presented here.
In addition to the lack of knowledge of plant host spectrum per site, I think there is a general lack of resolution for host bee diversity in their analysis of pathogen transmission. I understand that the Andrena community analysed in this study was composed of at least 4 distinct species. Did authors analyse the difference in AnBV1 prevalence and abundance across these 4 species? One can speculate that higher abundance of AnBV1 can be observed in one of them. In consequence, the prevalence and abundance of the virus may vary across sites, due to the changes in Andrena community composition (itself driven by change in landscape and/or flower provision). Did authors analyse the relative abundance of the 4 Andrena species across sites, and their foraging spectrum? May this explain the variation in AnBV1 prevalence in honeybees? I also wanted to ask what other bee species were present there? We have no information on potential other bee host present in these landscapes, that could have had an impact too.
I am also unsure why authors decided to categorize flower diversity (high vs. low) where they could have actually use either the community composition, or simply the actual numerical value of their diversity index. Another criticism to this analysis, is that 25m*25m plot used for this study sound big enough, but probably not representative of the total foraging spectrum of mining bees (and of course, of honeybees that show much larger flight distances). Can authors describe the landscapes around? How similar landscapes were from one site to the other? How was the flower diversity outside these plots? In general, I found that landscapes in which this work was performed were not very well described. In addition, the distance between sites is not optimum for this study. In fact, for sites separated by less than 2 km (average value for this study actually), the probability that honeybees come from the same colonies is very high. Can authors test the effect of distance between sites on virus prevalence?
Finally, I was surprised that authors, after claiming that lower flower diversity was influencing disease transmission for AnBV1, did not comment on the fact that this effect was not observed for other viruses. Authors found many other viruses in honeybees, but seems to not be present in mining bees. Can authors discuss this? I understand, from the introduction, that mining bees can carry these diseases (other papers shown presence of DWV, LSV & BQCV in their Andrena samples). What makes that AnBV1 can be transmitted by low plant diversity habitats, but not other viruses.
Overall, I think that part of the ms is very well reported, and clearly deserve publication, while the section on transmission seems incomplete and deserve either more work, are to be more careful with its interpretation and acknowledge limitations due to lack of data.
Minor comments:
- In line 108, authors mentioned regional decline of Andrena. Can they provide a reference for this?
- In line 276, authors mentioned they screened for many viruses. Which ones? Which genomes used?
- In lines 287 to 291, authors mentioned diptera viruses were found in their samples? What was the level of sequence identity? Were they really present in bees, or are they new viruses (i.e. not yet described)?
- I was confused with the sequence analysis of genomes that is supposed to demonstrate transmission. For example, BQCV sequences from honeybees and mining bees showed 96% identity. Likewise, in RbSh site, a 92% identity score was found for DWV between bee taxa. Is that not enough to say they diverge quite a bit? If transmission happens often, then one would expect higher similarity, isn’t it? Otherwise, to show sign of transmission, I think this section would deserve an analysis in greater depth, with an analysis considering substitution rates of viruses, and phylogenetic trees, showing how close are the viral variants in one location from different host species. As is it, I did not find this analysis very convincing.
- In general, I found the section 2.2 long and difficult to read. I would suggest to simplify it.
- Line 368, the sentence seems unfinished: central protein more similar to which type of DWV? Also, is the consensus for DWV is a true recombinant virus present in your sample, or a chimera produced by the assembly?
- How successful the virus enrichment was? I could not come across a number that demonstrates how relevant the treatment of nucleic was.
- Line 567, authors say that differences in RNA1 and RNA2 segments of AnBV1 qPCR could be due to qPCR efficiencies. Can authors actually test this? They calculated PCR efficiencies, they can correct their result with it, I think.
- AnBV1 abundance was varying a lot in mining bees, but not in honeybees. Is it possible that presence of AnBV1 in honeybee is driven by viral load in mining bees?
- I was confused by figure 2B and its caption. I don’t think I understood how to read it.
- Line 738: R2c is the conditional R2 value. What is the R2m value for the model?
Other comments:
- I really enjoyed the fact that experimental AnBV1 infection in honeybee cells were performed. That clearly adds to the manuscript. It’s a nice demonstration.
- I also liked that authors stressed the point of designing their own primers based on the specific sequences found in their samples rather than relying on previously published sequences.
Author Response

(The authors gave the same response as above.)

Reviewer 4 Report
Manuscript “Metatranscriptome analysis of sympatric bee species identifies bee virus variants and a new virus, Andrena associated bee virus-1” provides new knowledge on the field of honeybee viruses using NGS technology. Research is very well designed and straightforward but the manuscript is too long. It can be improved by removing too much unnecessary details in the method section and removing repetitive parts in results and discussion. The content is interesting, opens an important view on bee viruses and their diversity in different environments. The manuscript has enough scientific merit to be published with minor revision on formatting and style of writing.
Author Response

(The authors gave the same response as above.)

Round 2
Reviewer 3 Report
I have carefully read the response to reviewers’ comments. I appreciate the effort from authors to address and clarify many points in their manuscript, such as the composition of the Andrena community, the composition of landscapes around the study plots, and the potential for autocorrelation between adjacent and geographically close sites. I also appreciate that authors explained in more details which variable was used for the model (i.e. floral diversity used by honeybees) and the addition of fig. 6A from supp material. All that are nice additions, that I am sure will make the manuscript clearer for readers.
However, I am still thinking that a piece of information is missing there, about the intensity of overlap of flower use between mining bees and honey bees. The new figure 6A shows indeed that in low diversity plots, honey bees forage on a lower number of flower species, whilst the abundance of flowers shared with mining bees is high. From that, and with the support of their model (figure 6C), authors conclude that virus transmission is driven by low flower diversity, assuming that on low diversity habitats, most honeybees were foraging on the same flowers to mining bees. But what the data is not showing, is the quantitative measure of floral share by mining bees and honey bees. Honeybees may indeed forage on the same flowers that mining bees on low diversity habitats, but what was the proportion of honeybees foraging on the shared flower species? The impact of shared resource on disease transmission will not be the same if, let say, out of 100 honey bees, one was observed on the shared resource with Andrena, and 99 were seen foraging on another flower (not shared) vs. 90 honey bees on shared flowers, and 10 on non-shared. The quantitative measure of shared resource (or niche overlap as I called in my first round or review) is what is missing in this paper. This is derived for the plant-pollinator networks that is not available here -this is also something I stressed in my first review: the detail of plant-insect visitation per site were not given. In other words, we don’t know the proportion of honeybees sharing the same resource with mining bees in low diversity habitats. That’s why I keep thinking that without this information, direct transmission of AnBV1 between mining bees to honeybees is speculative -especially as the other insect visitors in these habitats were not screened for the virus. If only few honeybees were visiting the shared resource in low diversity habitats, figure 6A would still be valid, but I would doubt that the virus is directly transmitted from mining bees to honeybees. I think authors should either add the quantitative data, or at least discuss the caveat of not looking at the proportion of honeybees using the same resource.
As for the figure legend I could not understand clearly, that was not figure 2B, nor 3B, but figure 6B (sorry for the mistake in the first round).